# A model for regulation by SynGAP-α1 of binding of synaptic proteins to PDZ-domain 'Slots' in the postsynaptic density

Ward G Walkup IV[1], Tara L Mastro[1], Leslie T Schenker[1], Jost Vielmetter[2], Rebecca Hu[1], Ariella Iancu[1], Meera Reghunathan[1], Barry Dylan Bannon[1], Mary B Kennedy[1]*

[1]Division of Biology and Biological Engineering, California Institute of Technology, Pasadena, United States; [2]Beckman Institute Protein Expression Center, California Institute of Technology, Pasadena, United States

**Abstract** SynGAP is a Ras/Rap GTPase-activating protein (GAP) that is a major constituent of postsynaptic densities (PSDs) from mammalian forebrain. Its α1 isoform binds to all three PDZ (PSD-95, Discs-large, ZO-1) domains of PSD-95, the principal PSD scaffold, and can occupy as many as 15% of these PDZ domains. We present evidence that synGAP-α1 regulates the composition of the PSD by restricting binding to the PDZ domains of PSD-95. We show that phosphorylation by $Ca^{2+}$/calmodulin-dependent protein kinase II (CaMKII) and Polo-like kinase-2 (PLK2) decreases its affinity for the PDZ domains by several fold, which would free PDZ domains for occupancy by other proteins. Finally, we show that three critical postsynaptic signaling proteins that bind to the PDZ domains of PSD-95 are present in higher concentration in PSDs isolated from mice with a heterozygous deletion of synGAP.

*For correspondence: kennedym@its.caltech.edu

## Introduction

We propose a new model for regulation of trapping of AMPARs and other synaptic proteins in 'slots' at the postsynaptic membrane. Numerous experiments have shown that AMPARs become immobilized at the postsynaptic membrane by a three stage process involving insertion of receptors into the perisynaptic membrane, diffusion into the synapse and 'trapping' within the structure of the postsynaptic density (PSD; *Opazo and Choquet, 2011*). The sites at which the receptors are trapped have been referred to as 'slots,' (*Shi et al., 2001*) and they are believed to consist principally of PDZ domains on PSD-95 (*Opazo et al., 2012*). Phosphorylation by CaMKII of TARPs (Transmembrane AMPAR Regulatory Proteins; *Bredt and Nicoll, 2003*) increases their affinity for PDZ domains of PSD-95 suggesting that the phosphorylation may promote trapping of new AMPARs in the PSD (*Tomita et al., 2005*; *Opazo et al., 2010*). Indeed the PDZ domains of PSD-95 act as docking sites for several synaptic regulatory proteins (*Figures 1* and *2B*); including NMDA-type glutamate receptors (NMDARs; *Kornau et al., 1995*), as well as neuroligins (*Varoqueaux et al., 2006*) and LRRTMs (Leucine Rich Repeat TransMembrane proteins; *Linhoff et al., 2009*), which nucleate new synapse formation and contribute to clustering of AMPARs (*Siddiqui et al., 2010*). AMPARs, NMDARs, TARPs, LRRTMs, and neuroligins comprise the most highly enriched transmembrane proteins precipitated together with PSD-95 from the PSD fraction of mouse forebrain (*Dosemeci et al., 2007*).The multiplicity of PDZ binding proteins at the synapse raises the question of whether and how competition for binding to the PDZ domains is regulated at individual synapses.

SynGAP, a postsynaptic GTPase activating protein, is unusually abundant in the PSD scaffold. One prominent alternatively spliced isoform, synGAP-α1 (*Li et al., 2001*) contains a PDZ-domain

**eLife digest** The formation of memories is believed to depend on the strengthening of connections, called synapses, between neurons in the brain. When neurons are activated together, their synaptic connections become permanently strengthened to record the memory. This strengthening is called activity-dependent long-term potentiation.

As long-term potentiation develops, more protein receptors are added to the receiving side of the synapse. This allows the receiving neuron to produce a larger electrical response to the signaling chemicals it receives from the neuron on the sending side of the synapse. The addition of receptors is regulated by a set of enzymes held near the membrane of the synapse by a protein scaffold known as the postsynaptic density.

A major scaffold protein called PSD-95 contains binding sites, known as PDZ domains, that hold protein receptors and regulatory enzymes in place. One regulatory enzyme called synGAP is present in large numbers in the postsynaptic density and binds to the same PDZ domains as the receptors. Humans that have just one copy (instead of the usual two) of the gene that encodes synGAP have cognitive disabilities that are often accompanied by autism and epilepsy.

By studying purified proteins, Walkup et al. found that adding phosphate groups to synGAP reduces the enzyme's ability to bind to the PDZ domains. This reduced binding ability could make more PDZ domains available to bind to protein receptors and hold them at the synapse.

To measure the effect of reduced synGAP levels on the proteins found at postsynaptic densities, Walkup et al. used mice that had just one copy of the synGAP gene in their neurons. These mice have less synGAP in their postsynaptic densities and more of three proteins that bind to PDZ domains. These proteins hold receptors in the synapse and help synapses to form. Thus, synGAP may restrict the binding of other proteins to the PDZ domains in order to regulate the strength of the synapse.

Further experiments are now needed to investigate the importance of restriction by synGAP of binding to PDZ domains under a variety of circumstances in which the activity of neurons alters the strength of synapses.

ligand and binds to all three of the PDZ domains of PSD-95 (*Kim et al., 1998*; *Figures 1*, *2*). Here we propose that association of synGAP-α1 with PDZ domains of PSD-95 contributes to regulation of docking of AMPARs within the PSD and, therefore, to regulation of its overall protein composition. This function is distinct from its function as a Ras/Rap GAP (*Walkup et al., 2015*). We support this hypothesis by showing that phosphorylation by at least two protein kinases on several sites in the regulatory region of synGAP-α1 reduces the affinity of synGAP-α1's PDZ-ligand for the three PDZ domains of PSD-95. One of these protein kinases is CaMKII, which is activated when synaptic plasticity is initiated by activation of NMDA-type glutamate receptors (NMDARs). The other is PLK2, a constitutively active kinase that is induced by neuronal activity and mediates homeostatic scaling (*Seeburg et al., 2005*). We further show that binding of $Ca^{2+}$/CaM to synGAP-α1 selectively reduces its affinity for PDZ3 of PSD-95. Finally, we show that reduction of the total amount of synGAP in heterozygous knockout mice alters the composition of PSDs in the mutant mice in a way that is most directly explained by reduced competition from synGAP-α1 for binding to PDZ domains of PSD-95.

The PSD is an organized complex of signaling proteins attached to the postsynaptic membrane of excitatory glutamatergic synapses in the central nervous system. It comprises a network of scaffold proteins, the most prominent of which is PSD-95, a member of the MAGUK family (Membrane-Associated GUanylate Kinase-like proteins) (*Kennedy, 2000*; *Sheng and Kim, 2011*). An average PSD (~360 nm in diameter) is estimated to contain ~300 molecules of PSD-95 with 900 PDZ domains (*Chen et al., 2005*; *Sugiyama et al., 2005*). SynGAP is nearly as abundant in the PSD as PSD-95 itself (*Chen et al., 1998*; *Cheng et al., 2006*; *Dosemeci et al., 2007*). It is expressed as several isoforms with four different C-termini, α1, α2, β, and γ (*Li et al., 2001*; *McMahon et al., 2012*).The α1-containing isoforms are abundantly expressed at synapses and contain the PDZ ligand. Assuming that α1 isoforms make up 30–50% of the total synaptic synGAP, and that one synGAP-α1 molecule can bind to any one of the three PDZ domains in each molecule of PSD-95, synGAP-α1 could occupy

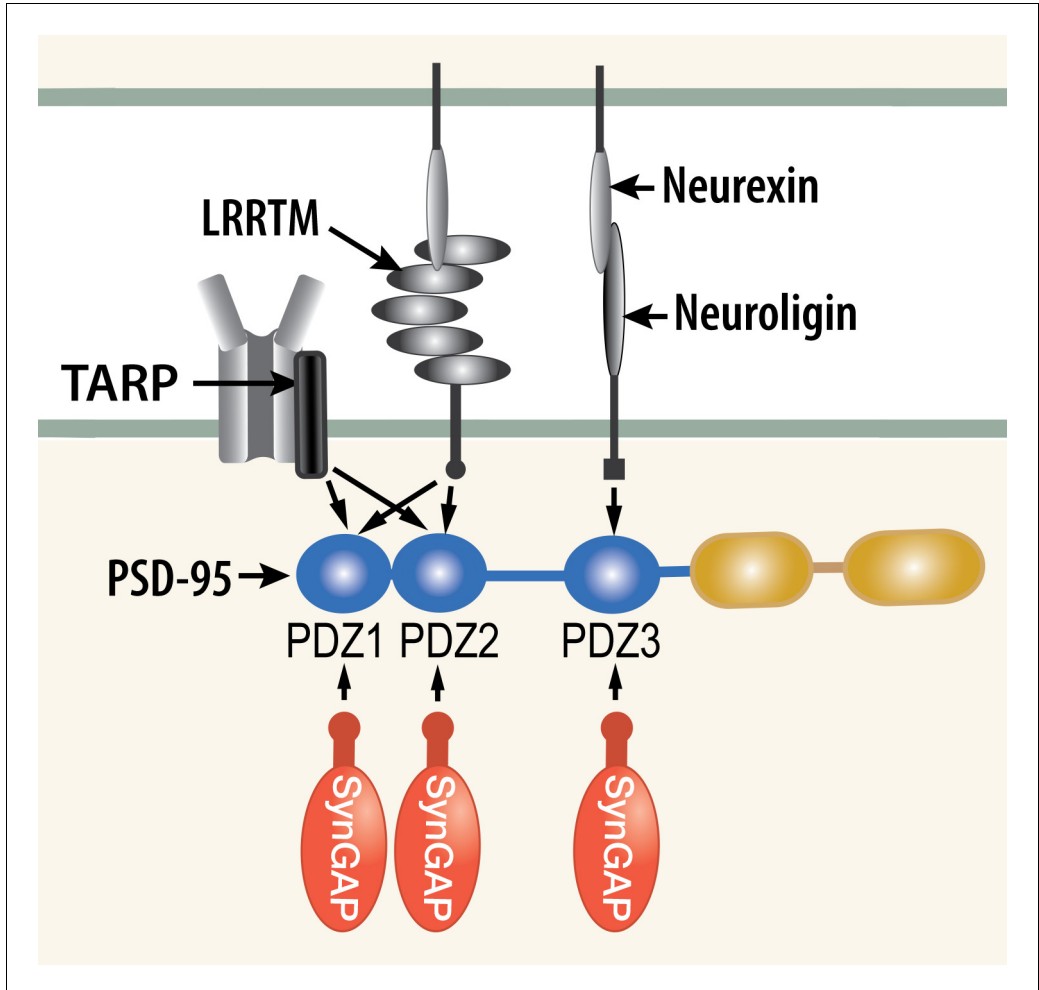

**Figure 1.** Competition among synaptic proteins for binding to PDZ domains of PSD-95. Each of the three PDZ domains of PSD-95 binds only one protein at a time. The composition of the PSD-95 complex is determined by a dynamic equilibrium that depends on the relative affinities of the proteins that bind to it and their relative concentrations. The figure illustrates competition among three prominent membrane proteins (TARP, LRRTM, and neuroligin) and cytosolic synGAP-α1. Note that LRRTMs and neuroligin also bind across the cleft to presynaptic neurexins. Other PSD proteins that bind to PDZ1 and PDZ2 include the GluN2A and GluN2B subunits of the NMDARs, plasma membrane $Ca^{2+}$ pumps (PMCA2B and 4B), the ErbB4 receptor (**Lim et al., 2002**), and BRAG1 (an ArfGEF; **Sakagami et al., 2008**; **Myers et al., 2012**). PDZ3 binds a smaller group of additional proteins, including β1-adrenergic receptors, and CRIPT (**Lim et al., 2002**).

10–15% of the PDZ domains of PSD-95 in an average PSD. Thus, it could help to regulate the size and strength of the synapse by limiting the availability of 'slots' that can bind AMPAR complexes (**Hayashi et al., 2000**; **Shi et al., 2001**; **Opazo et al., 2012**). This proposed function could explain its unusually high abundance in the PSD which, until now, has been mysterious (**Chen et al., 1998**; **Sheng and Hoogenraad, 2007**).

It has been proposed that, in general, PDZ domains act as flexible protein interaction points that can be modified to support changes in cytoplasmic organization (**Kurakin et al., 2007**). Complexes formed by PDZ domain interactions are examples of linked, multiple equilibria, the stable configurations of which are determined by the concentrations of each component and by their affinities for the relevant PDZ domains. Evidence has indicated that PSD-95 protein complexes exist in dynamic equilibrium permitting continual turnover and potential rearrangement of their composition (**Sturgill et al., 2009**; **Schapitz et al., 2010**). Because synGAP-α1 is an abundant protein in the PSD with a relatively high affinity for all three PDZ domains of PSD-95, reduction of its affinity after

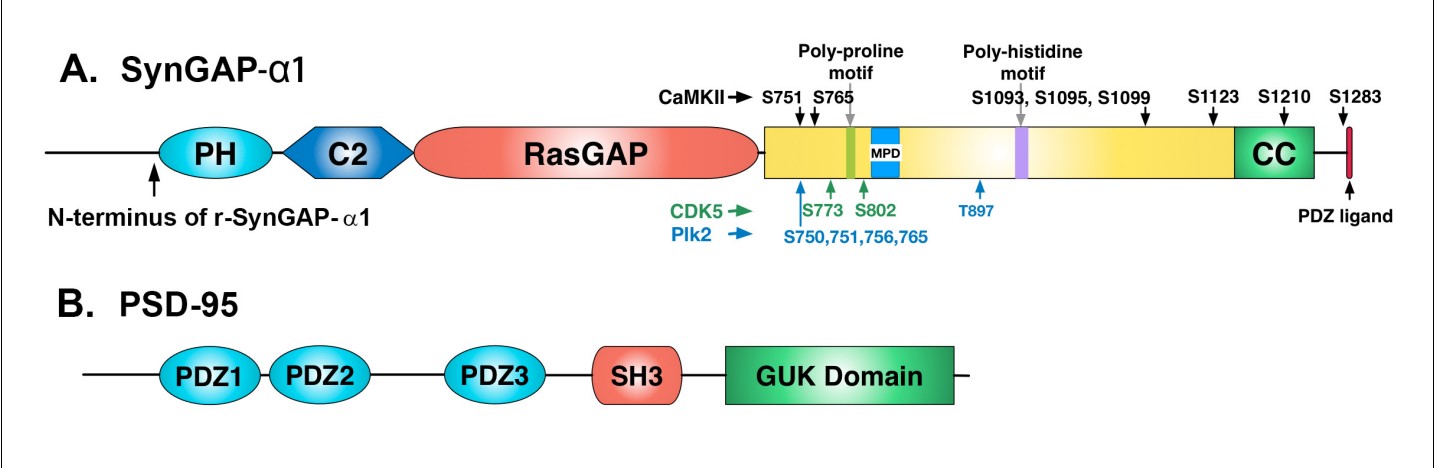

**Figure 2.** Domain diagrams of synGAP-α1 and PSD-95. (**A**) SynGAP-α1. The N-terminus of r-synGAP-α1 is indicated, as are the locations of the major sites phosphorylated by CaMKII (black), CDK5 (green) (*Walkup et al., 2015*), and PLK2 (blue) (Walkup IV et al.; *Lee et al., 2011*), most of which are in the 'disordered domain.' 'MPD' indicates a region in which several nearby serines (808, 810, 821, 825, and 827)are phosphorylated by both CaMKII and PLK2. Numbering is based on rat isoform synGAP A1-α1. For comparison, phosphorylation sites identified in *Lee et al., (2011)* were numbered according to the B isoform and can be compared to ours by subtracting 44 residues from our numbering. Phosphorylation sites identified in *Araki et al., (2015)* were numbered according to the A isoform and can be compared to ours by adding 15 residues to our numbering. The 5 residue PDZ ligand is located at the C-terminus. (**B**) PSD-95. The five major domains of PSD-95, including the approximate relationships of its three N-terminal PDZ domains are indicated (*Cho et al., 1992*).

phosphorylation will allow other components to compete more effectively for binding; thus, the composition of the PSD-95 complex will shift to a new equilibrium.

This proposed mechanism can account for many previous experimental observations. For example, two other groups used imaging to show that phosphorylation by CaMKII in living neurons triggers movement of synGAP away from the PSD (*Yang et al., 2013*; *Araki et al., 2015*). In the Araki study, the authors suggested that phosphorylation of synGAP results in 'dispersion' of synGAP away from the PSD and therefore has the effect of upregulating Ras near the PSD. We propose here that an additional important consequence of the decrease in binding of synGAP to the PDZ domains of PSD-95 is readjustment of the composition of the PSD resulting from increased availability of the PDZ domains of PSD-95. Indeed, the dynamics of movements of synGAP and AMPARs visualized in living neurons following synaptic stimulation are consistent with this hypothesis. Activation of NMDARs and CaMKII causes dispersal of synGAP away from the PSD within a few minutes (*Yang et al., 2013*; *Araki et al., 2015*). The same stimuli produce an equally rapid increase in the rate of trapping of AMPARs at synaptic sites (*Makino and Malinow, 2009*; *Opazo et al., 2010*; *Opazo et al., 2012*). Thus, the rates of these two processes, observed in living neurons, are compatible with the notion that reduced binding of synGAP to PSD-95 during induction of LTP opens up binding slots for AMPAR complexes. Dispersal of synGAP away from the postsynaptic membrane would be expected to result in increased activation of Ras and Rap, which would lead to increases in the rates of exocytosis and/or endocytosis of AMPARs, respectively (*Zhu et al., 2002*). However, experiments show that increased exocytosis of AMPARs does not contribute to their increased trapping in the minutes following synaptic stimulation (*Makino and Malinow, 2009*). Instead, exocytosis replenishes surface AMPARs in the dendrite and perisynaptic spine membrane, and this process occurs with a slower time course than enhanced trapping.

An additional example of a previous result that is explained by this model, is the observation that absence of synGAP in hippocampal neurons cultured from synGAP KO fetuses leads to accelerated ('precocious') maturation of spines, including early movement of PSD-95 into spine heads, and ultimately larger clusters of PSD-95 in individual synapses compared to *wt* neurons (*Vazquez et al., 2004*). In this study, expression of *wt*-synGAP-α1 in the mutant neurons rescued all of these phenotypes; however, expression of synGAP-α1 with a deletion of the five residue PDZ ligand (ΔSXV) failed to rescue any of the effects of synGAP deficiency on precocious maturation of spines. In fact,

expression of synGAPΔSXV caused an increase in the size of clusters of PSD-95 in spines compared to *wt* neurons. This failure to rescue the phenotypes was not a result of mislocalization of synGAP-α 1; synGAPΔSXV localized like *wt*-synGAP-α1 to developing spine heads. Instead, the data are consistent with the idea that synGAP-α1 normally competes with several proteins for binding to the PDZ domains of PSD-95, and thus limits the size of clusters of PSD-95 and its associated proteins, as well as their movement into spine heads (*Vazquez et al., 2004*).

Yet another result supporting the model is the finding of *McMahon et al. (2012)* that the α1 and α2 isoforms have markedly different effects on synaptic strength when expressed in primary cultures of forebrain neurons. For example, 12 to 36 hr after transfection of the neurons with the A1α1 isoform of synGAP, 73% of the neurons were 'silent', that is, had no miniature end-plate potentials (mEPSCs), compared with 11% of control neurons. In contrast, transfection with the A1α2 isoform, which differs from α1 only in its C-terminal 48 residues and lacks a PDZ ligand, had no effect on the proportion of silent neurons, which remained at 11%. Like *Vazquez et al., (2004)*, they found that the differential effect did not arise from mislocalization of the A1α2 isoform. Both isoforms of syn-GAP localized normally to dendrites and spines. The most straightforward explanation of this result is consistent with our hypothesis, which predicts that overexpression of the synGAP-α1 isoform bearing the PDZ ligand restricts binding of AMPAR auxiliary proteins to PDZ domains and thus interfers with their localization to the synaptic site.

Neurons cultured from synGAP deficient mice have been reported by several groups to have a higher average number of AMPARs at their synapses than *wt* neurons (*Kim et al., 2003*; *Vazquez et al., 2004*; *Rumbaugh et al., 2006*). The data presented here suggests that the increase in AMPARs in *Syngap*[+/-] mice may be a direct result of increased binding of TARPs and LRRTMs to PDZ domains that are made available by the reduced amount of synGAP (*Tomita et al., 2005*; *de Wit et al., 2009*).

Finally, our proposed model and supporting results may help to explain the mechanism underlying a form of developmental intellectual disability (ID) resulting from synGAP haploinsufficiency. Mutations in a single copy of synGAP have been causally implicated in sporadic cases of non-syndromic ID, often associated with either autism (ASD) or epilepsy (*Berryer et al., 2013*). The frequency of developmental ID worldwide is estimated at 1 to 3%, and 25 to 50% of cases are sporadic, meaning that the parents are not affected. Although data is still sparse, mutations causing SynGAP haploinsufficiency appear to account for 2–9% of sporadic cases (*Hamdan et al., 2011*; *Berryer et al., 2013*), suggesting that its prevalence in the population could be as high as 0.03–0.1% and placing it in the same range of frequency as Fragile-X syndrome. The amount of synGAP in the brains of mice with synGAP haploinsufficiency is reduced by 50% (*Vazquez et al., 2004*). We show here that this reduction leads to a shift in the composition of the PSD scaffold, apparently resulting from the decrease in synGAP's ability to compete for binding to PDZ domains of PSD-95. This derangement is likely a significant factor in the human pathology leading to ID, ASD, and epilepsy.

## Results

### Phosphorylation of r-synGAP-α1 by CaMKII and PLK2 reduces its binding to PDZ domains of PSD-95

SynGAP-α1 can be expressed in bacteria and purified in a soluble form by deleting the first 102 residues of its N-terminus (*Walkup et al., 2015*). This version of synGAP, termed r-synGAP-α1, retains all of the identified functional domains, the regulatory domain, and the C-terminal PDZ ligand (*Figure 2A*). In a previous study we showed that r-synGAP-α1 is phosphorylated by CaMKII at several residues including S1283, which is 7 residues upstream of the PDZ domain ligand located at residues 1290–1293 (*Walkup et al., 2015*). Because this phosphorylation site is so near the PDZ ligand, we wondered whether its phosphorylation, or phosphorylation of other sites by CaMKII, would interfere with binding of synGAP-α1 to PDZ domains of PSD-95. To test this, we incubated r-synGAP-α1 with affinity resins substituted with recombinant PDZ domains as described under Materials and methods. The beads contained PDZ1, PDZ2, PDZ3, a fragment containing PDZ1 and PDZ2 (PDZ12), or a fragment containing all three PDZ domains (PDZ123) (*Figure 3—figure supplement 1*; *Walkup and Kennedy, 2014*). Binding of r-synGAP-α1 to the beads was tested with or without a

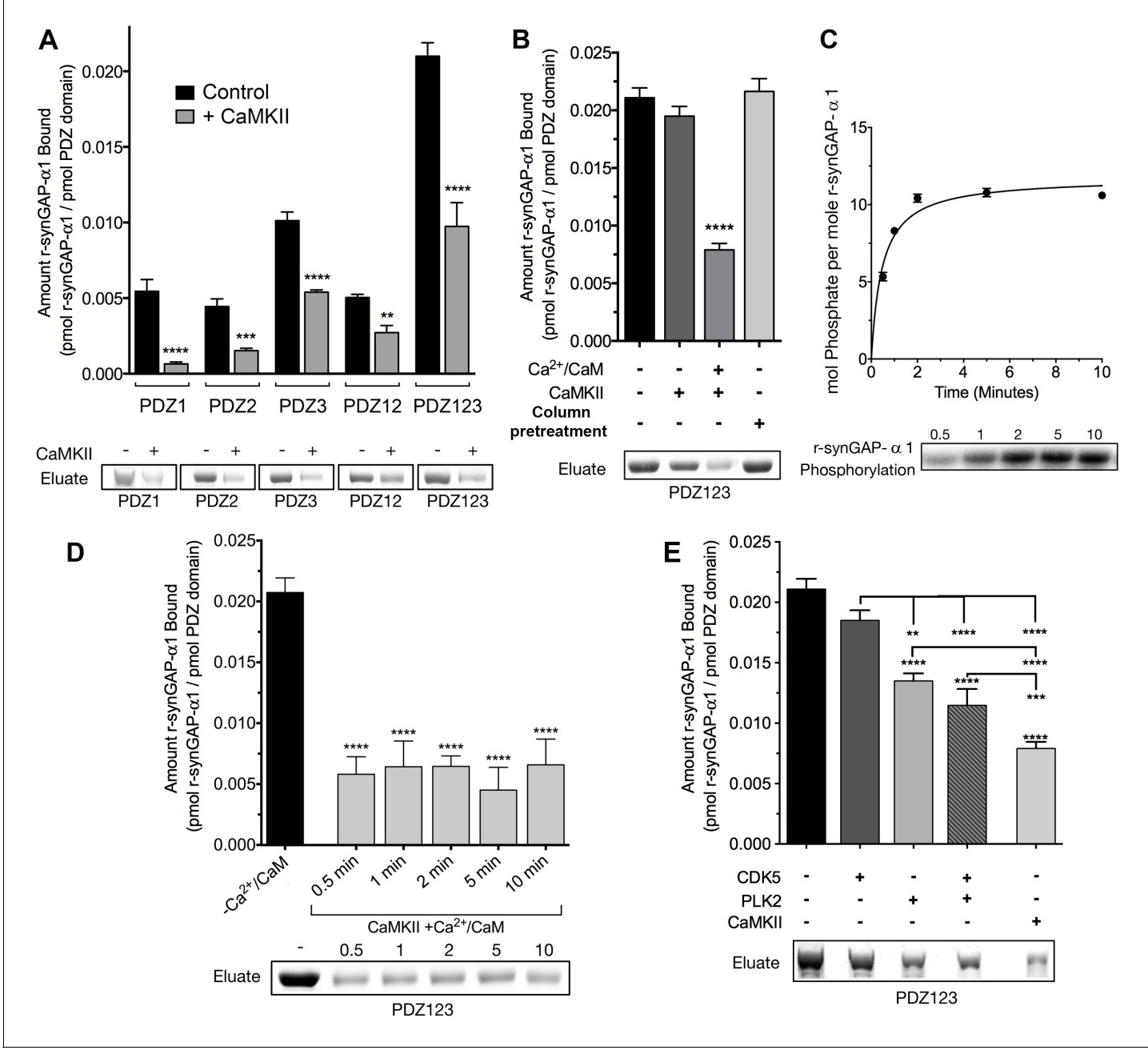

**Figure 3.** Phosphorylation by CaMKII regulates association of r-synGAP-α1 with PDZ domains of PSD-95. (**A**) Association of r-synGAP-α1 with PDZ domains of PSD-95 before and after phosphorylation by CaMKII. R-synGAP-α1 was incubated in a phosphorylation mix for 10 min with either 0 CaMKII and 0 Ca²⁺/CaM (control) or 10 nM CaMKII and 0.7 mM CaCl₂ /3.4 μM CaM (+ CaMKII) before binding to PDZ domain resins for 60 min at 25°C, as described under 'Materials and methods.' There is no detectable binding of synGAP to unsubstituted resin, and no detectable non-specific binding of proteins to the PDZ resins under these conditions (*Walkup and Kennedy, 2014*). (**B**) Both Ca²⁺/CaM and CaMKII are required in the phosphorylation reaction to reduce binding of r-synGAP-α1 to PDZ123 resin. R-synGAP-α1 was incubated in the phosphorylation reaction without either Ca²⁺/CaM or CaMKII or with both before binding to PDZ resin. The final bar shows that phosphorylation of the PDZ123 domain resin itself doesn't alter binding of r-synGAP-α1. PDZ123 domain affinity resin was phosphorylated for 60 min in the presence of CaMKII and 0.7 mM CaCl₂/3.4 μM CaM before incubation with control r-synGAP (500 nM) for 60 min at 25°C. (**C**) Stoichiometry of phosphorylation of r-synGAP-α1 by CaMKII. R-synGAP-α1 (725 nM) was phosphorylated in the presence of CaMKII (10 nM), as described under 'Materials and methods.' At the indicated times, reactions were quenched by addition of 3x Laemmli sample buffer. Radiolabeled r-synGAP-α1 was isolated by SDS-PAGE and quantified as described under 'Materials and methods.' (**D**) Change in affinity of r-synGAP-α1 for PDZ123 after phosphorylation by CaMKII for times corresponding to those measured in *C*. R-synGAP-α1 was phosphorylated for 0.5 to 10 min as described in *C* before incubation with PDZ123 domain affinity resin for 60 min as described under 'Materials and methods.' Control (-CaMKII, -Ca²⁺/CaM) is r-synGAP-α1 incubated in the phosphorylation reaction in the absence of CaMKII and

*Figure 3 continued on next page*

*Figure 3 continued*

Ca$^{2+}$/CaM. (**E**) Change in affinity of r-synGAP-α1 for PDZ123 after phosphorylation for 10 min by CDK5 or PLK2. or by a combination of the two, as described in 'Materials and methods.' The reduction in binding after phosphorylation for 10 min by CaMKII is shown for comparison. Data shown in **A-E** are plotted as mean ± S.E. (n = 4). For A, B, D, and E, the statistical significance of differences in binding to PDZ domain resin relative to unphosphorylated r-synGAP-α1 control (-Ca$^{2+}$/CaM) was determined by ordinary one way ANOVA (uncorrected Fisher's LSD). **p<0.01; ***p<0.001; ****p<0.0001.

The following source data and figure supplement are available for figure 3:

**Source data 1.** Source data for *Figure 3A*.

**Figure supplement 1.** Purification of recombinant PDZ domains of PSD-95.

prior 10 min phosphorylation by CaMKII. As expected, without phosphorylation, r-synGAP-α1 binds specifically to each of the three PDZ domains (*Figure 3A*). In this assay, its binding is highest to PDZ3. Binding of r-synGAP-α1 to PDZ123 reveals a substantial avidity effect; that is, the amount bound per individual PDZ domain is twice that bound to PDZ3 alone and four times that bound to either PDZ1 or PDZ2 alone.

Phosphorylation by CaMKII reduces binding of r-synGAP-α1 to all of the individual PDZ domains and to PDZ12 and PDZ123 (*Figure 3A*). The reduction in binding requires the presence of both Ca$^{2+}$/CaM and CaMKII in the phosphorylation reaction mixture (*Figure 3B*). The fourth bar of *Figure 3B* shows that the reduction in binding is not caused by phosphorylation of PDZ domains on the column by residual CaMKII. We have shown previously that as many as 10 sites on r-synGAP-α1 are phosphorylated by CaMKII (*Walkup et al., 2015*). Approximately 5 of these, including site S1123, are fully phosphorylated after an 0.5 min reaction (*Walkup et al., 2015* and *Figure 3C*). To test whether the reduction in binding depends primarily on phosphorylation of the rapidly phosphorylated sites or requires phosphorylation of most of the sites, we tested binding of r-synGAP-α1 to PDZ123 after phosphorylation for various times (*Figure 3D*). The reduction in binding is maximal after 0.5 min, indicating that phosphorylation of the more rapidly phosphorylated sites is sufficient for full reduction of affinity.

R-synGAP-α1 can also be phosphorylated by CDK5 (*Walkup et al., 2015*) and by PLK2 (*Lee et al., 2011*) both of which increase its RasGAP activity. We tested the effect of 10 min of phosphorylation by each of these kinases on affinity of r-synGAP-α1 for PDZ123. Phosphorylation by CDK5 had no significant effect on affinity; whereas, phosphorylation by PLK2 decreased its affinity by ~40%, less strongly than does phosphorylation by CaMKII (*Figure 3E*). Both CaMKII and PLK2 can phosphorylate several sites in a region marked MPD (multiple-phosphorylation domain) in *Figure 2A*, including serines at 808, 810, 821, 825, and 827 (*Walkup et al., 2015*; Walkup et al., in preperation). This result means that phosphorylation of multiple sites at various locations in the regulatory region can decrease binding of r-synGAP-α1 to PDZ domains, but to varying degrees.

## Affinity of r-synGAP-α1 for the PDZ domains of PSD-95 determined by surface plasmon resonance (SPR)

The three PDZ domains of PSD-95 are located in the N-terminal half of the protein from residues 61 to 402. The first two PDZ domains are separated by 4 residues and the third is 53 residues downstream. We determined the affinities of r-synGAP-α1 for individual PDZ domains, for PDZ12 (residues 61–249) and for PDZ123 (residues 61–402) by a competition assay in which SPR is used to detect the amount of free r-synGAP-α1 in solutions containing a constant amount of r-synGAP-α1 and varying amounts of recombinant PDZ domains (*Nieba et al., 1996*; *Lazar et al., 2006*; *Abdiche et al., 2008*). To detect the free r-synGAP-α1, recombinant PDZ domains are immobilized on the Biacore chip as described under 'Materials and methods.' We used the competition method rather than conventional Biacore measurements in which varying concentrations of r-synGAP-α1 are applied to a chip containing immobilized PDZ domains because concentrations of r-synGAP-α1 above ~100 nM produced a large bulk resonance signal caused by high viscosity that obscured the change in resonance produced by its binding to PDZ domains. The competition assay eliminates the need to apply high concentrations of r-synGAP-α1 to the chip.

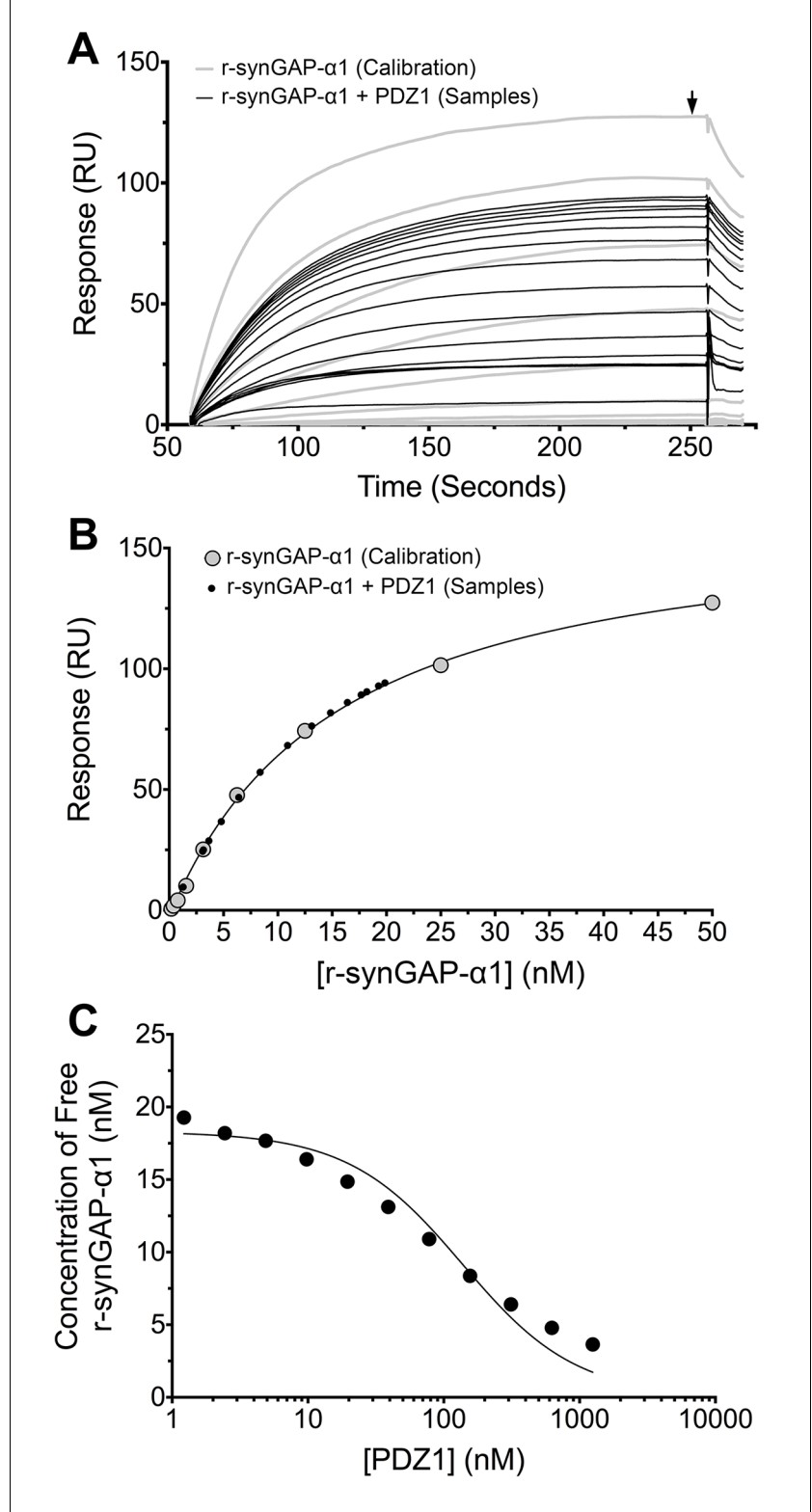

**Figure 4.** Measurement of affinity of r-synGAP-α1 for PDZ1 of PSD-95 by the 'competition in solution' method. (**A**) Biacore sensorgrams showing the calibration curves (grey lines) for binding of 0–50 nM r-synGAP-α1 and the measurement of free r-synGAP-α1 (samples; black lines) in mixtures containing 25 nM r-synGAP-α1 and 0–10 μM PDZ1 domain. Free r-synGAP-α1 was detected by binding to PDZ1 domains immobilized on a Biacore chip as described under 'Materials and methods.' (**B**) A standard calibration curve was constructed by plotting the

*Figure 4 continued on next page*

*Figure 4 continued*

maximum calibrated resonance responses (marked by arrow in *A*) against the corresponding concentrations of r-synGAP-$\alpha$1 (large grey dots). The maximum resonance responses for each sample mixture were plotted on the standard curve to determine the free r-synGAP-$\alpha$1 concentrations in each mixture (black dots). (*C*) Plot of free r-synGAP-$\alpha$1 concentrations determined in *B* against the log of PDZ domain concentrations (black circles). The data were fit to the binding equation shown in 'Materials and methods' with the use of Biacore software. A $K_D$ value (*Table 1*) was calculated from the equation as described under 'Materials and methods.'

We generated a standard curve in which the maximum resonance responses of a series of concentrations of r-synGAP-$\alpha$1 from 0 nM to 50 nM (*Figure 4A*, grey traces) were determined and plotted against r-synGAP-$\alpha$1 concentration (*Figure 4B*, large grey dots). The data were fit with a hyperbolic curve. The maximum resonance response of a series of mixtures containing 25 nM r-synGAP-$\alpha$1 and increasing concentrations of PDZ1 from 0 nM to 10 $\mu$M were measured, and the concentration of r-synGAP-$\alpha$1 remaining free to bind to PDZ1 on the chip was then determined from the standard curve (*Figure 4B*, small black dots). A $K_D$ of 220 $\pm$ 30 nM (*Table 1*) was calculated as described under Materials and methods (*Figure 4C*). We used the same method to measure $K_D$s for PDZ2 and PDZ3 (*Figure 5A and B*, respectively). Because, PDZ12 and PDZ123 contain more than a single PDZ domain binding site, all of which have similar affinities, we determined single 'apparent' equilibrium dissociation constants ($K_{Dapp}$) for these two constructs. As discussed in 'Materials and methods', it is not possible to derive a unique equation incorporating two or three binding sites when the affinities of the multiple sites are similar. The best fit of the data with the equation for a single binding site provides a $K_{Dapp}$ that can be used to characterize the binding behavior of the constructs (*Figure 5C and D*, respectively). The values are summarized in *Table 1*. We obtained an additional value of 730 $\pm$ 50 nM for the $K_D$ of PDZ3 by a conventional Biacore assay, which is in good agreement with the $K_D$ measured by the competition assay. These data show that, under these conditions, PDZ1 has a higher affinity for r-synGAP-$\alpha$1 than does PDZ3.

## Binding of r-synGAP-$\alpha$1 phosphosite mutants to PDZ123 affinity resin

To examine which phosphorylation sites can reduce binding of r-synGAP-$\alpha$1 to PDZ domains, we measured binding of recombinant mutants of r-synGAP-$\alpha$1 to PDZ123 affinity resin. We first

**Table 1.** Affinities of R-synGAP-$\alpha$1 for PDZ Domains of PSD-95. Dissociation constants ($K_D$) and apparent dissociation constants ($K_{Dapp}$) for the interactions of r-synGAP-$\alpha$1 with the PDZ domains of PSD-95 were determined by the Biacore/SPR 'competition in solution' method as described under 'Materials and methods.' In one experiment, the $K_D$ for PDZ3 was determined by conventional SPR as described under 'Materials and methods.' Goodness of Fit refers to the fit of data shown in **Figures 4**, **5,** and **7** to the equation relating synGAP$_{free}$ to PDZ domain concentration, assuming a single binding site, as described under 'Materials and methods.' Because PDZ12, and PDZ123 contain more than one binding site for r-synGAP-$\alpha$1, the affinities are given as apparent dissociation constants. Data are expressed as mean $\pm$ S.E.

| PDZ Domain from PSD-95 | No. of Experiments | Dissociation Constant ($K_D$) for Binding to R-synGAP (nM) | Goodness of Fit ($R^2$) |
|---|---|---|---|
| PDZ1 | 3 | 220 $\pm$ 30 | 0.908 - 0.947 |
| PDZ2 | 2 | 1500 $\pm$ 100 | 0.967, 0.969 |
| PDZ3 | 2 | 620 $\pm$ 70 | 0.951, 0.962 |
| PDZ3 | 1 | 730 $\pm$ 50 (by conventional SPR) | N.A. |
| | | Apparent Dissociation Constant ($K_{Dapp}$) for Binding to R-synGAP (nM) | |
| PDZ12 | 4 | 350 $\pm$ 40 | 0.931 - 0.987 |
| PDZ123 | 7 | 4.7 $\pm$ 0.6 | 0.957 - 0.987 |
| PDZ123 | 3 | (r-synGAP Phosphorylated by CaMKII) 46 $\pm$ 10 | 0.810 - 0.880 |
| PDZ123 | 2 | (r-synGAP S1283D) 16 $\pm$ 3 | 0.953, 0.954 |

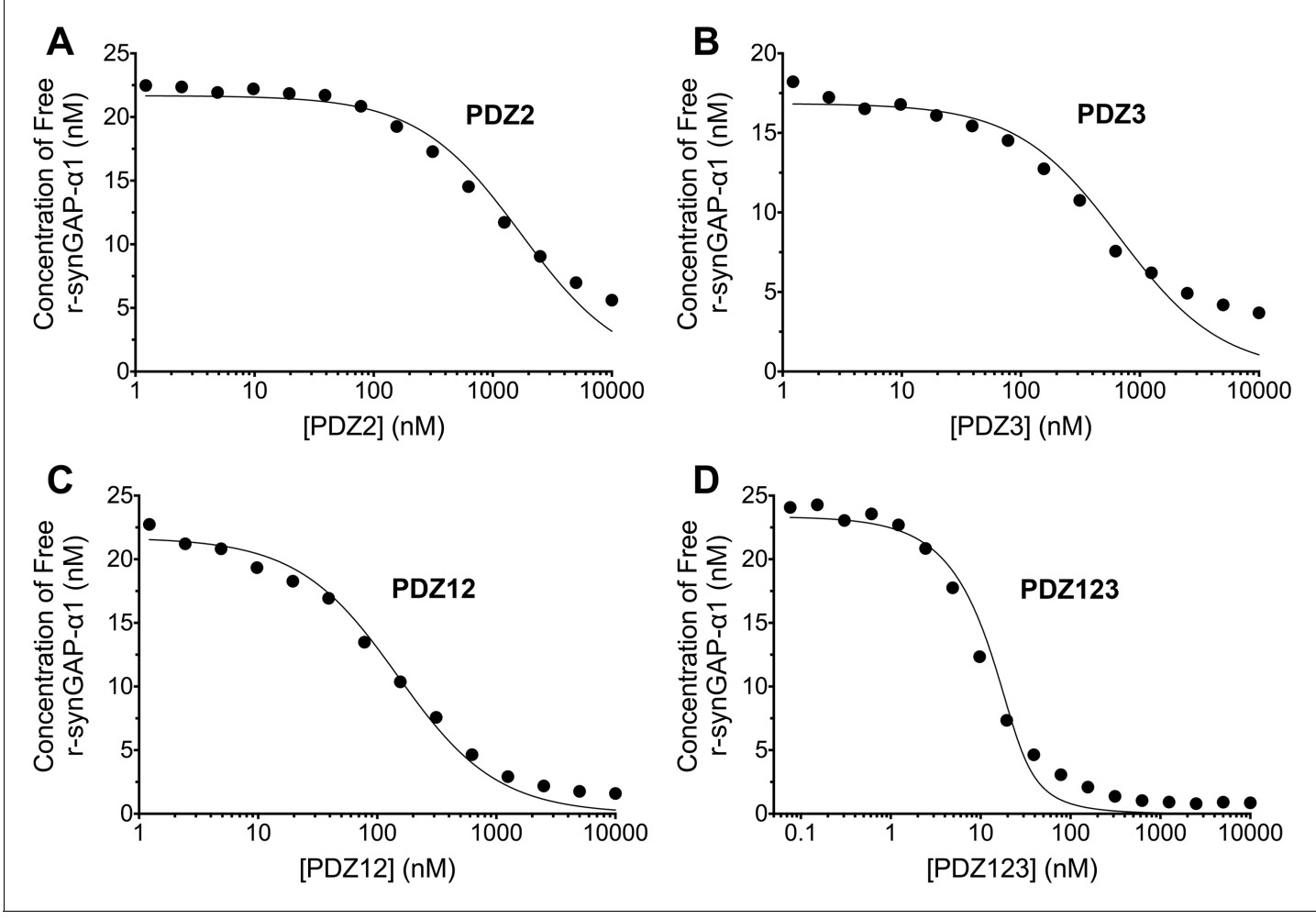

**Figure 5.** Affinities and apparent affinities of r-synGAP for PDZ2, PDZ3, PDZ12 and PDZ123 domains of PSD-95 determined by the 'competition in solution' method. The concentrations of free r-synGAP-α1 in sample mixtures containing each of the indicated PDZ domains were measured as described in *Figure 4A and B*, and under 'Materials and methods.' The values (black dots) were plotted against the log of the PDZ domain concentration and fit to a binding curve as described in *Figure 4C*. Representative experiment for **A**, PDZ2; **B**, PDZ3; **C**, PDZ12; and **D**, PDZ123. The calculated $K_D$ and $K_{Dapp}$ values from all experiments are listed in *Table 1*.

compared binding of all of the unphosphorylated mutants by ANOVA corrected for multiple comparisons by the Tukey method (*Figure 6*, black bars). Single or multiple mutations of the indicated serines to alanine did not significantly alter binding before phosphorylation; nor did the double mutation S1093D/S1123D. Only mutations in which site S1283 was mutated to the phosphomimetic aspartate (S1283D and S1093D/S1123D/S1283D) decreased binding of unphosphorylated r-synGAP-α1 to PDZ123 with a significant p-value (p<0.0001 for both). This result suggests that phosphorylation of S1283 has the most potent effect of all the sites on binding to PDZ domains. Mutations S1123A and S1283A did not significantly alter the change in binding after a brief, 0.5 min phosphorylation. However, they both produced a trend toward lower reduction of binding, suggesting that both sites contribute to reduction of binding after brief phosphorylation. The effect of the double mutation S1093A/S1123A after 0.5 min of phosphorylation was not significantly different from the effect of S1123A, suggesting that phosphorylation of S1093 doesn't influence binding to PDZ domains after a brief phosphorylation. In contrast, the triple mutant S1093A/S1123A/S1283A was the only S to A mutation that significantly interfered with the loss of binding (p=0.0074) compared to wild type after 0.5 min of phosphorylation. This result reinforces the conclusion that sites S1123

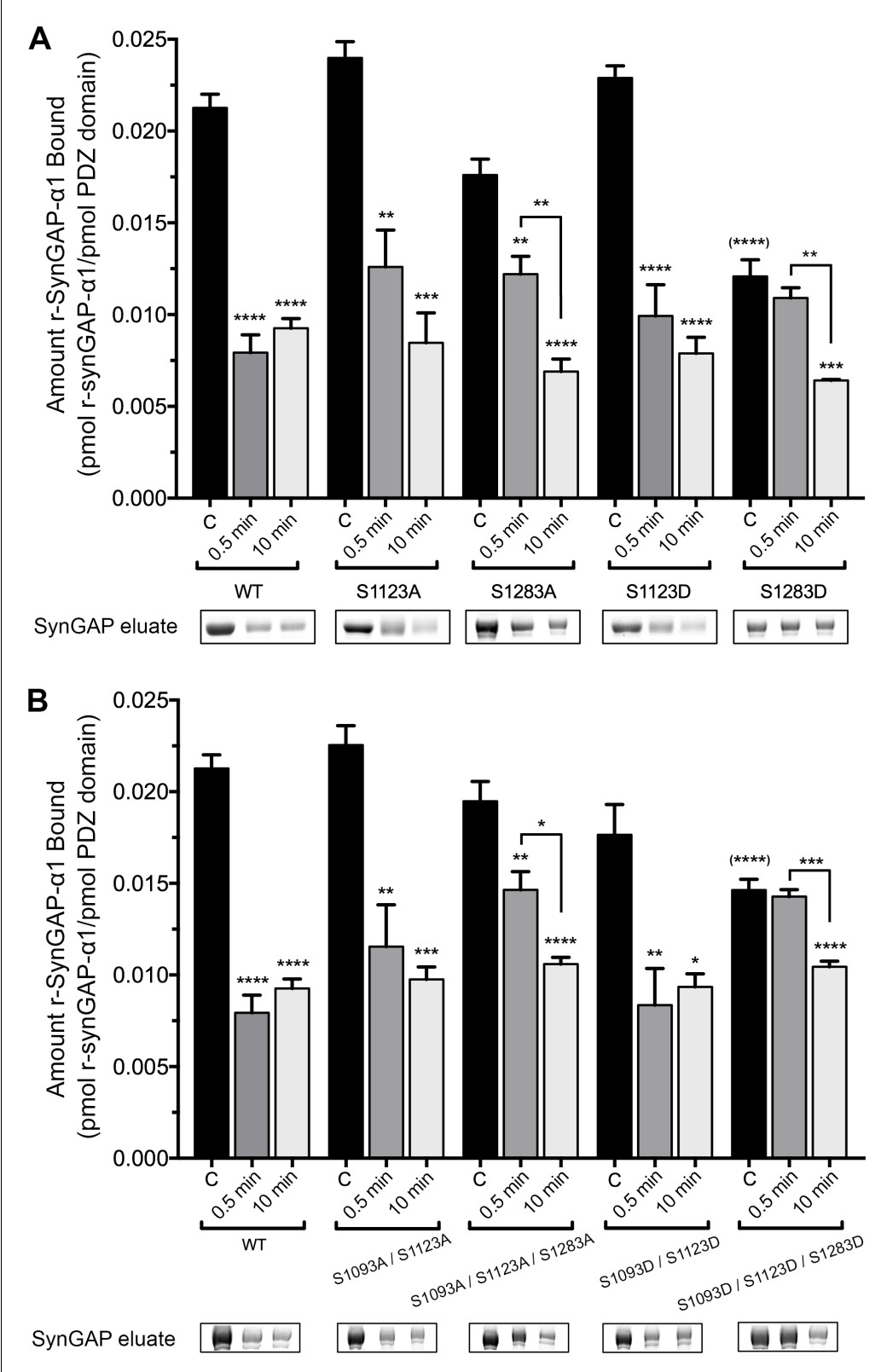

**Figure 6.** Effect of phosphorylation by CaMKII on association of PDZ123 domains with phospho-deficient and phospho-mimetic mutants of r-synGAP-α 1. Wild-type and mutant r-synGAP-α1 were incubated with phosphorylation mixtures for 10 min without (Control), or for 0.5 or 10 min with 10 nM CaMKII, 0.7 mM $CaCl_2$/3.4 µM CaM(CaMKII), then incubated with PDZ123 affinity resin for 60 min at 25°C, as described under 'Materials and methods.' (A) Binding to PDZ123 of r-synGAP-α1 and single mutations at S1123 and S1283. Sites S1123 or S1283 were mutated either to alanine (S1123A, S1283A)

*Figure 6 continued on next page*

Figure 6 continued

or to the phosphomimetic aspartate (S1123D, S1283D). (B) Binding to PDZ123 of r-synGAP-α1 and double or triple mutations at S1093, S1123, and S1283, as indicated. Data are mean ± S.E. (n = 4). The statistical significances of differences between wild type and the various mutants in binding to PDZ123 before phosphorylation were determined by ordinary one way ANOVA (Tukey correction for multiple comparisons). Only mutations including S1283D were significantly different from wild type (indicated by (****)). Differences between unphosphorylated and phosphorylated individual mutants were compared individually by ordinary one way ANOVA (uncorrected Fisher's LSD). *pp<0.05; **p<0.01; ***p<0.005; ****p<0.0001. See the text for additional statistical comparisons.

and S1283 both contribute to reduction of binding of r-synGAP-α1 to PDZ domains after brief phosphorylation.

Notably, none of the mutations interfere with the loss of binding to PDZ domains after ten min of phosphorylation (*Figure 6A and B*). Taken together, these results and the results of phosphorylation by PLK2 mean that phosphorylation of S1283 and S1123 significantly reduces binding of r-synGAP-α1 to PDZ domains, however maximum loss of binding can also be accomplished by cumulative phosphorylation over ten min of several sites within the regulatory domain (*Figure 2A* and *Walkup et al., 2015*). This mechanism is consistent with the finding of Holt et al. that clusters of sites in rapidly evolving disordered regions appear to shift position in evolutionary time suggesting that regulation by phosphorylation often involves disruption or enhancement of protein-protein interactions by addition of multiple negative charges (*Holt et al., 2009*).

## Effect of phosphorylation on affinity of r-synGAP-α1 for PDZ123 measured by SPR

We measured the apparent dissociation constants ($K_{Dapp}$'s) for binding to PDZ123 of r-synGAP-α1 phosphorylated for 10 min by CaMKII, and of the phosphomimetic mutant r-synGAP-α1 S1283D, as described for *Figure 5* and under 'Materials and methods'. Phosphorylation for 10 min by CaMKII increases the $K_{Dapp}$ of r-synGAP-α1 approximately ten-fold (*Figure 7A*, *Table 1*); whereas mutation of S1283 to aspartate increases the $K_{Dapp}$ approximately four-fold (*Figure 7B*, *Table 1*). Thus, cumulative phosphorylation of several residues can reduce affinity for PDZ domains by an order of magnitude; whereas, addition of a negative charge at S1283 alone can reduce the affinity by a factor of four.

## $Ca^{2+}$/CaM binds directly to r-synGAP-α1

While studying phosphorylation of r-synGAP-α1 by CDK5 (*Walkup et al., 2015*), we found that the presence of $Ca^{2+}$/CaM in reactions with either CDK5/p35 or CDK5/p25 doubled the rate and stoichiometry of the phosphorylation (*Figure 8A and B*). Inclusion of $Ca^{2+}$ or CaM alone in the phosphorylation reactions did not alter the rates or stoichiometry.

We tested whether this effect resulted from binding of $Ca^{2+}$/CaM to CDK5/p35 (e.g. *He et al., 2008*), or CDK5/p25 by comparing the rates of phosphorylation of histone H1, a well-known substrate of CDK5 in the presence and absence of $Ca^{2+}$/CaM (*Figure 8C and D*). Phosphorylation of histone H1 by either CDK5/p35 or CDK5/p25 was unaffected by $Ca^{2+}$/CaM. This result suggests that $Ca^{2+}$/CaM binds directly to r-synGAP-α1, causing a substrate-directed enhancement of its phosphorylation.

To further verify that $Ca^{2+}$/CaM binds directly to r-synGAP-α1, we showed that r-synGAP-α1 binds to a CaM-Sepharose affinity resin in a $Ca^{2+}$-dependent manner, as would be expected if it binds $Ca^{2+}$/CaM specifically and with significant affinity (*Figure 8—figure supplement 1*). We found that the presence of $Ca^{2+}$/CaM alone in a Ras- or Rap-GAP assay has no effect on the GAP activity of r-synGAP-α1 (*Figure 8—figure supplement 2*).

## Affinity of binding of R-synGAP-α1 to $Ca^{2+}$/CaM

We measured the affinity of binding of $Ca^{2+}$/CaM to r-synGAP-α1 by the conventional SPR method on the Biacore as described under Materials and methods. CaM was immobilized on a chip, and r-synGAP-α1 was applied to it at concentrations from 0 to 75 nM (*Figure 9A*). Analysis of the equilibrium phase of association at each concentration (*Figure 9B*) yielded a $K_D$ of 9 ± 1 nM, indicative of high affinity binding.

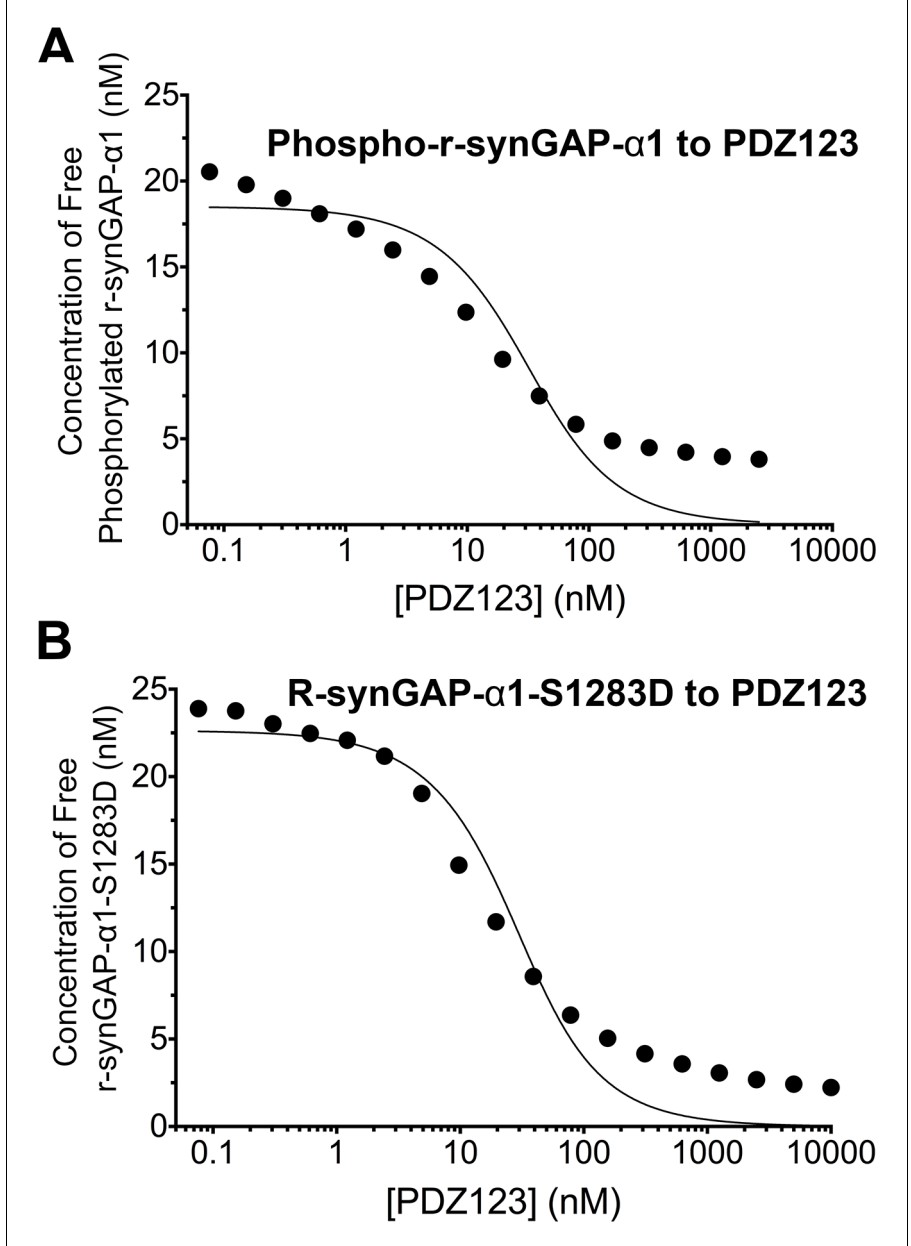

**Figure 7.** Apparent affinities of phosphorylated r-synGAP-α1 and r-synGAP-α1-S1283D for PDZ123 determined by the 'competition in solution' method. Representative plots of the concentrations of (**A**) free phospho-r-synGAP-α1 phosphorylated as described for PDZ Binding Assays under 'Materials and methods,' and (**B**) r-synGAP-α1-S1283D, measured in sample mixtures containing PDZ123 as described in *Figure 4A and B*, and under 'Materials and methods.' The values (black dots) were plotted against the log of the PDZ123 concentration in the mixture and fit to a binding curve as described in *Figure 4C*. The calculated values of $K_{Dapp}$ are listed in *Table 1*.

To begin to define the location of the high affinity $Ca^{2+}$/CaM binding site, we compared the affinities for $Ca^{2+}$/CaM of r-synGAP-α1 and a C-terminal truncated protein, sr-synGAP (residues 103–725) by a bead-binding assay as described under Materials and methods. We measured the amount of each protein bound to a fixed amount of CaM-Sepharose after incubation with increasing concentrations (*Figure 9C and D*). Both r-synGAP-α1 (*Figure 9C*) and sr-synGAP (*Figure 9D*) showed saturable binding to the CaM-Sepharose resin, and did not bind to control Sepharose beads lacking CaM (data not shown). The data were fit to hyperbolic curves and the $K_D$'s for binding of r-synGAP-

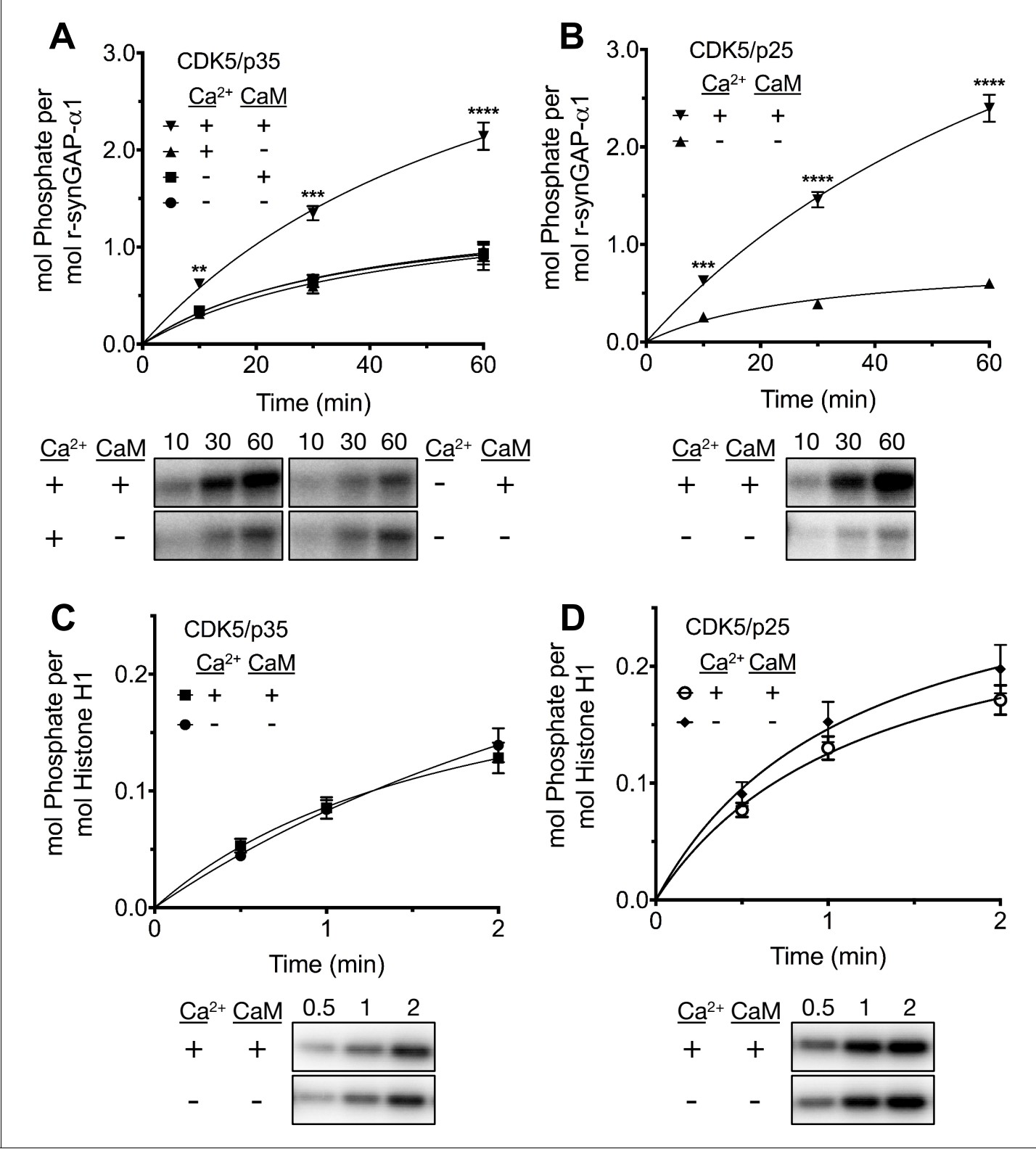

**Figure 8.** Effect of Ca$^{2+}$/CaM on stoichiometry of phosphorylation of r-synGAP-α1 and histone H1 by CDK5. Stoichiometry of phosphorylation of r-synGAP-α1 (**A** and **B**) and histone H1 (**C** and **D**) by CDK5/p35 or CDK5/p25. R-synGAP-α1 (286 nM) or histone H1 (4.3 μM) were incubated with CDK5/p35 or CDK5/p25 as described under 'Materials and methods' in the presence or absence of 0.7 mM CaCl$_2$ or 3.4 μM CaM, as indicated in each panel. Reactions were quenched at the indicated times by addition of 3x Laemmli sample buffer and radiolabeled r-synGAP-α1 and histone H1 were

*Figure 8 continued on next page*

*Figure 8 continued*

quantified as described under 'Materials and methods.' Data are plotted as mean ± S.E. (n = 4–7). The statistical significance of differences in phosphorylation in the presence of $Ca^{2+}$ and CaM were determined by ordinary one way ANOVA (uncorrected Fisher's LSD). **p<0.01; ***p<0.001; ****p<0.0001.

The following figure supplements are available for figure 8:

**Figure supplement 1.** R-synGAP-α1 binds to CaM affinity resin.

**Figure supplement 2.** Effect of $Ca^{2+}$/CaM on GAP activity of r-synGAP-α1.

α1 and sr-synGAP to $Ca^{2+}$/CaM were calculated to be 31 ± 3 nM and 210 ± 30 nM, respectively. Thus, the high affinity site appears to be located in the regulatory disordered region of r-synGAP-α1, which is missing in sr-synGAP. The $K_D$'s determined by the bead-binding assay (31 ± 3 nM) and Biacore equilibrium binding (9 ± 1 nM) are in the range of those reported for calcineurin (PP2B) and CaMKII, 1–10 nM (*Hubbard and Klee, 1987*; *Cohen and Klee, 1988*) and 40–80 nM (*Miller and Kennedy, 1985*; *Meyer et al., 1992*; *Hudmon and Schulman, 2002*), respectively. We did not detect any binding when sr-synGAP was injected onto the CaM-substituted Biacore chip at concentrations from 10–2500 nM. Thus, the relatively weak binding of sr-synGAP observed in the bead-binding assay is not reproducible when measured on the Biacore chip. These data indicate that $Ca^{2+}$/CaM binds only weakly, if at all, to the N-terminal half of synGAP. A meta-analysis algorithm for detecting potential CaM-binding domains (*Mruk et al., 2014*) predicts two $Ca^{2+}$/CaM binding sites in the C-terminal half of r-synGAP-α1, one from residues 1000–1030 and another in the putative coiled coil domain from residues 1229–1253. The SPR measurements do not allow us to confirm or to rule out the presence of two high affinity sites of similar affinity.

## Effect of $Ca^{2+}$/CaM on binding of r-synGAP-α1 to PDZ domains of PSD-95

We tested whether binding of $Ca^{2+}$/CaM alters the binding of r-synGAP-α1 to PDZ domains by comparing binding to each affinity resin in the presence or absence of $Ca^{2+}$/CaM (*Figure 10A*). The presence of $Ca^{2+}$/CaM during incubation with resin significantly reduces binding of r-synGAP-α1 to PDZ3 and to PDZ123, but not to PDZ1 and/or PDZ2. Thus, binding of $Ca^{2+}$/CaM has a more specific, but weaker, effect on binding to the PDZ domains than does phosphorylation. The effects of phosphorylation and of the presence of $Ca^{2+}$/CaM during incubation with resin are not additive (*Figure 10B*); that is, the presence of $Ca^{2+}$/CaM during the incubation with resin does not further decrease binding of phosphorylated r-synGAP-α1 to PDZ123.

## SynGAP haploinsufficiency alters the composition of the PSD

The physiological significance of the finding that phosphorylation by CaMKII decreases the affinity of r-synGAP-α1 for the PDZ domains of PSD-95 is best considered in the context of the high copy number of synGAP in the PSD. In molar terms, synGAP-α1 is 30–50% as abundant in the PSD as PSD-95 itself (*Chen et al., 1998*; *Cheng et al., 2006*; *Dosemeci et al., 2007*; *Sheng and Hoogenraad, 2007*). Our data suggest that phosphorylation of synGAP-α1 by CaMKII, triggered by activation of NMDARs, would promote dissociation of synGAP-α1 from the PDZ domains, reducing its ability to compete with other proteins for binding.

*Syngap$^{+/-}$* mice have been shown to have about half as much synGAP in homogenates from forebrain as *wt* littermates. Because binding equilibria are driven not only by the intrinsic affinities of the binding partners, but also by their concentrations, one prediction of our proposed hypothesis is that synGAP haploinsufficiency, which reduces the amount of synGAP in the brain by 50% (*Vazquez et al., 2004*), will cause a significant increase in binding to PSD-95 of other prominent PSD-95 binding proteins, such as TARPs, LRRTMs, or neuroligins. Thus, PSDs isolated from *Syngap$^{+/-}$* mice would be predicted to have less synGAP and more TARPs, LRRTMs, and/or neuroligins bound to PSD-95 than do PSDs isolated from *wt* mice. We prepared PSD fractions from the forebrains of six *Syngap$^{+/-}$* mice and from six *wt* litter mates and measured the ratios of synGAP, TARPs, LRRTM2,

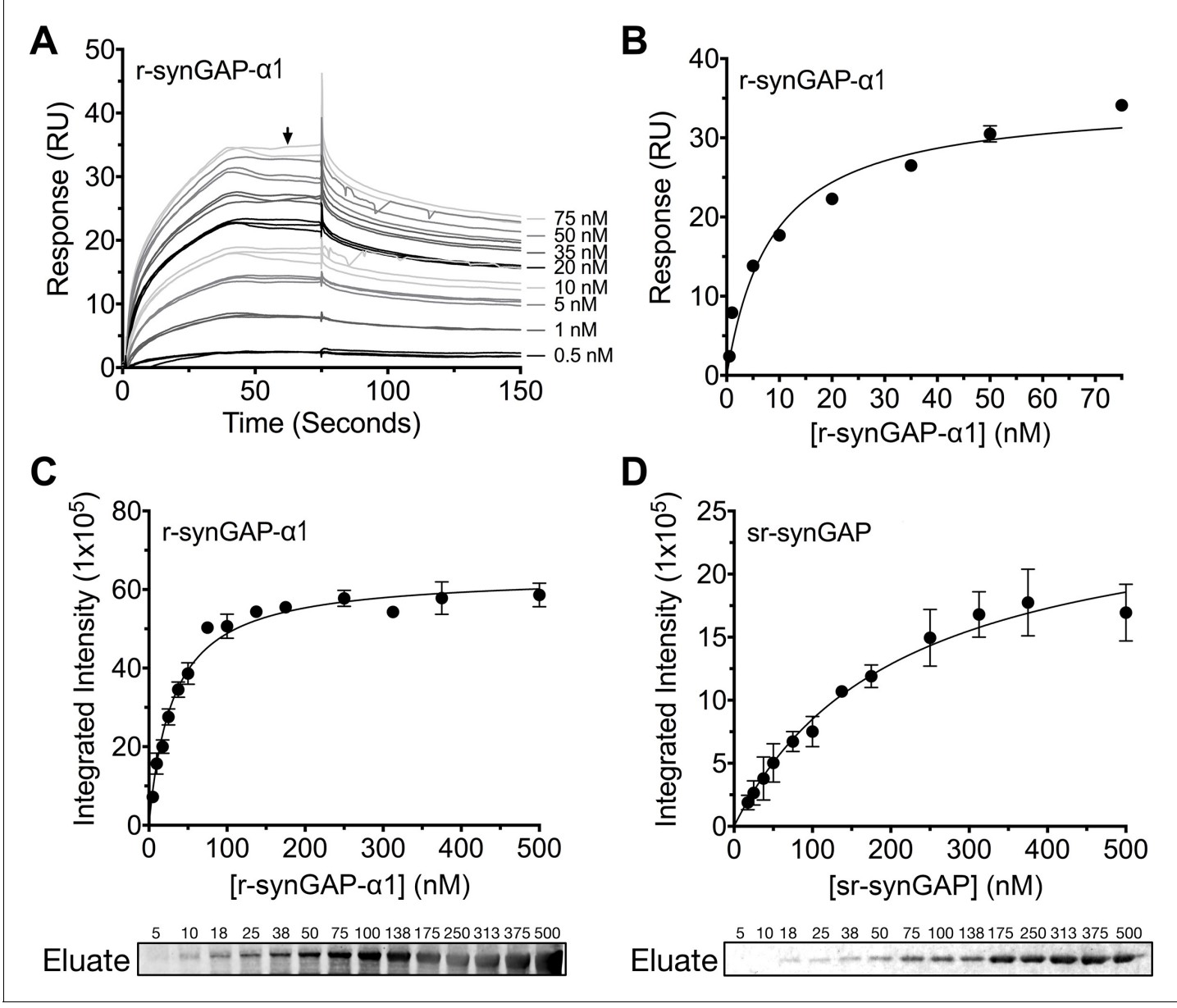

**Figure 9.** Affinity of r-synGAP-α1 and sr-synGAP for $Ca^{2+}$/CaM determined by equilibrium analysis. (**A** and **B**) The affinity of r-synGAP-α1 for $Ca^{2+}$/CaM was measured by SPR with CaM immobilized on the chip and r-synGAP-α1 injected at 0–75 nM onto the chip surface as described under 'Materials and methods.' (**A**) Sensorgrams with the blank and reference flow cell readings subtracted show the response upon injection of r-synGAP-α1 onto the chip surface (0–75 s) and its dissociation from the chip surface (75–150 s). (**B**) RUs at equilibrium (marked by arrow in **A**) were plotted against the corresponding concentrations of r-synGAP-α1 and fitted to a hyperbolic curve. A $K_D$ of 9 ± 1 nM was calculated as described under 'Materials and methods.' (**C** and **D**) The affinities of r-synGAP-α1 (**C**) and sr-synGAP (**D**), (0–500 nM) for $Ca^{2+}$/CaM were measured by incubation with CaM-Sepharose resin as described under 'Materials and methods.' Integrated intensities of bound r-synGAP-α1 and sr-synGAP were measured from immunoblots as described under 'Materials and methods' and plotted versus the corresponding concentrations incubated with resin. Integrated intensities from Western blots were linear over the range of r-synGAP-α1 and sr-synGAP concentrations used in the assays. Data in *C* and *D* are plotted as mean ± S.E. (n = 3).

neuroligin-1, and neuroligin-2 to PSD-95 in the two fractions by quantitative immunoblot as described in 'Materials and methods' (*Figure 11*). The recoveries of total protein and the amount of PSD-95 per µg protein were identical in the two preparations. As predicted, the level of synGAP is decreased in relation to PSD-95 by ~ 24% in PSDs (p=0.0007, d = 1.75) from the *Syngap*[+/-] mice (*Figure 11A*). Furthermore, the ratios of TARPs 2,3,4,γ8, and of LRRTM2 to PSD-95 are significantly

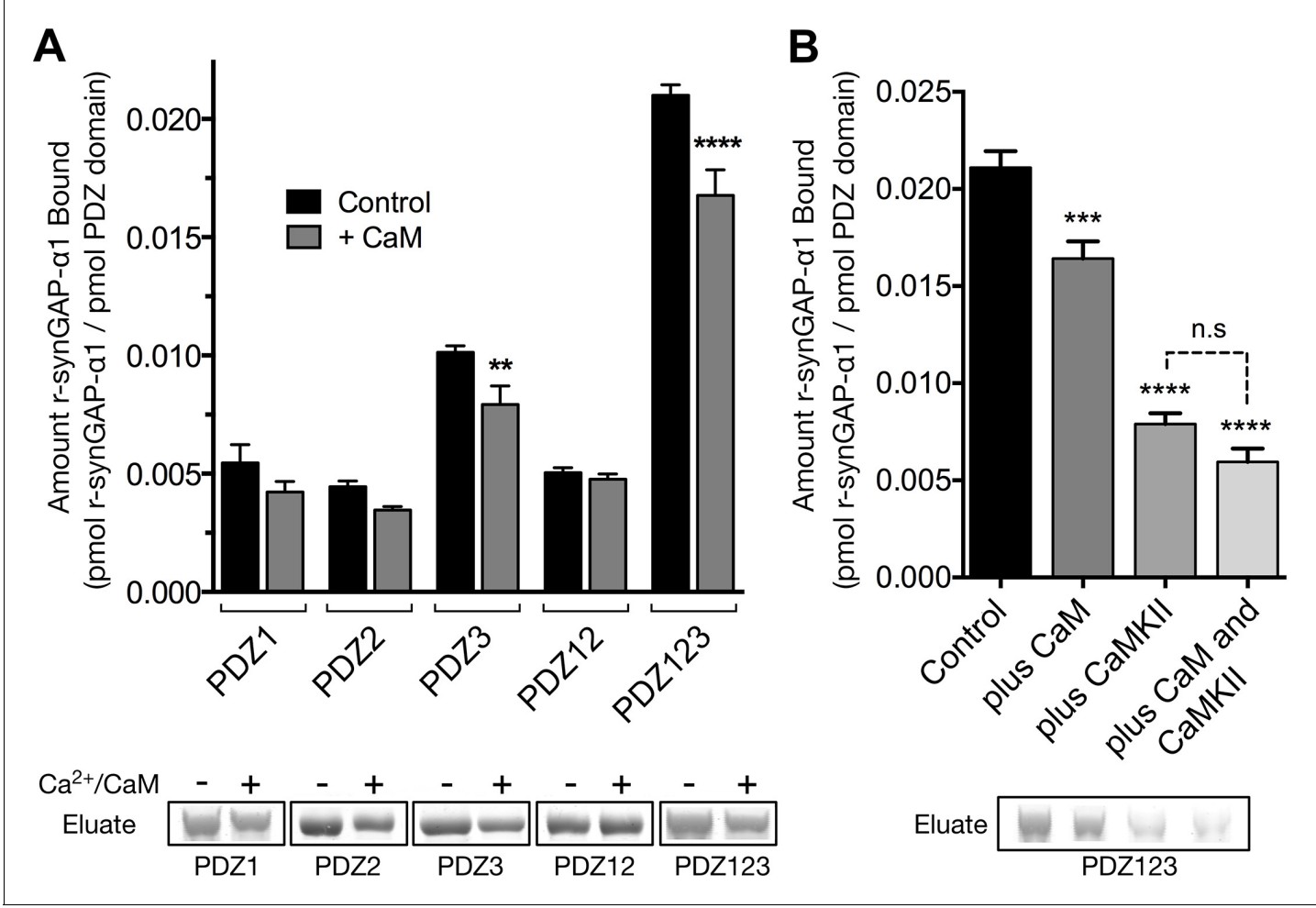

**Figure 10.** Effect of Ca²⁺/CaM binding on association of r-synGAP-α1 with PDZ domains of PSD-95. (**A**) Association of control and Ca²⁺/CaM bound r-synGAP-α1 with PDZ domains of PSD-95. R-synGAP-α1 (500 nM) without (Control) or with (+ CaM) 0.7 mM CaCl₂/3.4 μM CaM was incubated with PDZ domain resins (PDZ1, PDZ2, PDZ3, PDZ12, and PDZ123) for 60 min at 25°C and bound r-synGAP-α1 was measured as described under 'Materials and methods.' (**B**) Effects of bound Ca²⁺/CaM and phosphorylation by CaMKII on association of r-synGAP-α1 with PDZ123 domain are not additive. The association with PDZ123 domain resin of Control, Ca²⁺/CaM bound (plus CaM), phosphorylated r-synGAP-α1 (plus CaMKII) and phosphorylated r-synGAP-α1 bound to Ca²⁺/CaM (plus CaM and CaMKII) was measured as described in A. Data are plotted as mean ± S.E. (n = 4). The statistical significance of differences in PDZ domain binding relative to Control was determined by ordinary one way ANOVA (uncorrected Fisher's LSD). **p<0.01; ****p<0.0001.

The following source data is available for figure 10:

**Source data 1.** Source data for *Figure 10A*.

increased (*Figure 11B,C*; TARP/PSD-95, ~12%, p=0.017, d = 0.93; LRRTM2/PSD-95, ~14%, p=0.0035, d = 0.66). This result strongly suggests that, as predicted, the increase in availability of PDZ1/2 domains on PSD-95 in the *Syngap⁺/⁻* mice enhances steady-state binding of TARPs and LRRTMs to those sites. Interestingly, the ratio of neuroligin-1 to PSD-95 is unchanged in the *Syngap⁺/⁻* mice (d = 0.07, *Figure 11D*), suggesting that increased availability of PDZ3 on PSD-95 is not a strong driver of association of neuroligin-1 with the PSD fraction. However, the ratio of neuroligin-2 to PSD-95 (*Figure 11E*) is increased by ~9% (p=0.019, d = 0.64). Neuroligin-2 normally associates mostly with inhibitory synapses and mediates their maturation (*Varoqueaux et al., 2004*). However, *Levinson et al. (2005)* reported that over-expression of PSD-95 in neurons causes a redistribution of neuroligin-2, increasing the proportion associated with excitatory synapses. Thus, the effect of reduction of synGAP on the distribution of neuroligin-2 shown here is the same as the effect

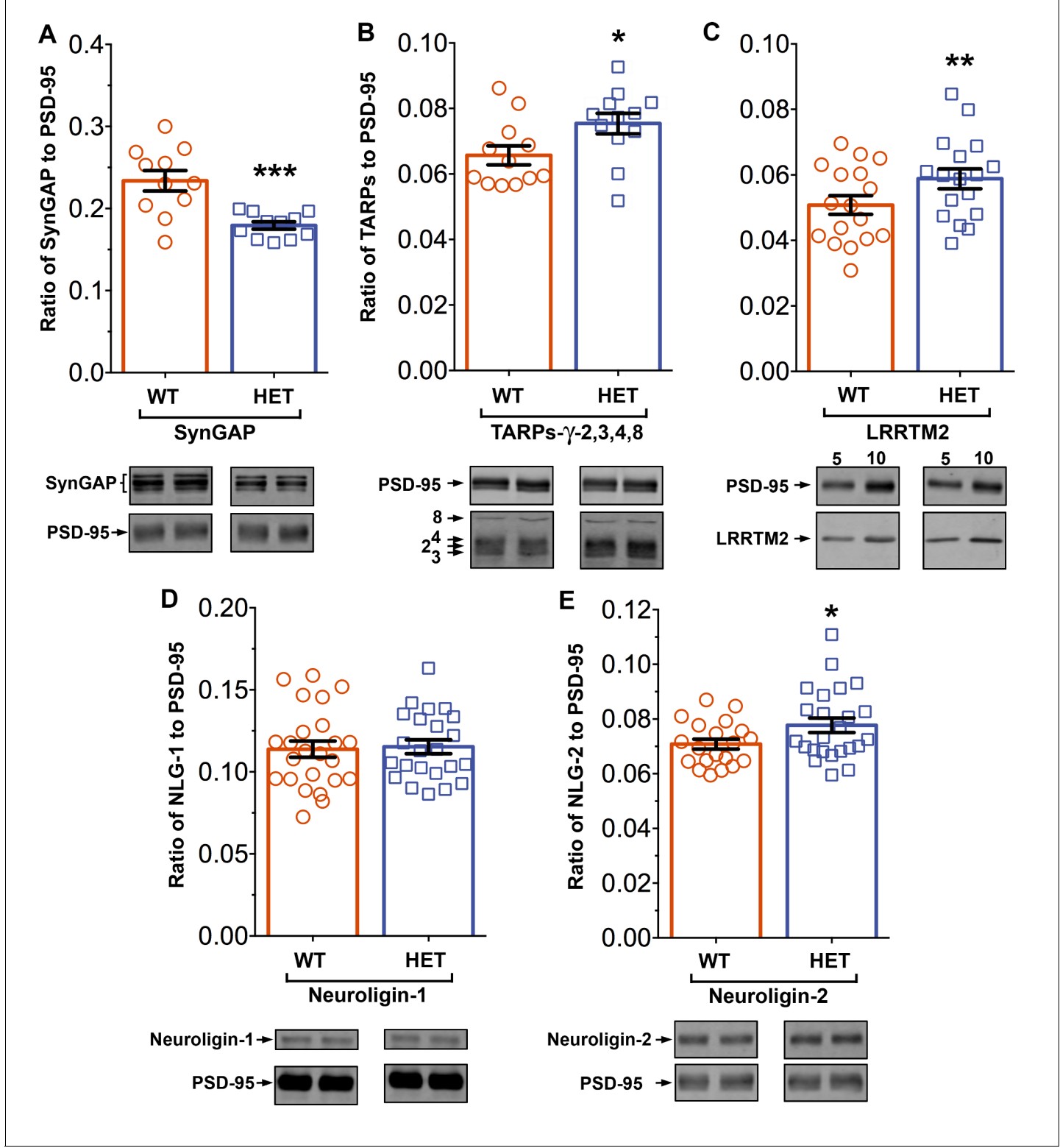

**Figure 11.** Altered composition of the PSD in mice with heterozygous deletion of *synGAP*. Ratios of amounts of the indicated proteins to PSD-95 in each lane were measured as described in 'Materials and methods' and are reported as mean ± S.E. For all blots except those for neuroligin-1, PSD-95 was detected with a secondary Ab labeled with AlexaFluor680 and the binding protein was detected with secondary Ab labeled with IRDye 800. On the neuroligin-1 blot, both PSD-95 and neuroligin-1 were detected with AlexaFluor680; the two bands were well-separated in each lane. Representative sets of visualized bands for WT and HET from the same blot are shown below the graphs. Individual points represent the ratio of the indicated protein to

*Figure 11 continued on next page*

*Figure 11 continued*

PSD-95 in a single lane (n refers to these technical replicates). The WT and HET PSD preparations were each made from six animals; thus, each of the two preparations represents six biological replicates. (**A**) SynGAP to PSD-95 ratio. Data were collected for 22 lanes from two blots containing 5 μg total PSD fraction per lane. One blot contained six lanes WT and six lanes HET samples, the other contained five lanes of each. The mean ratio of synGAP to PSD-95 was 0.234 ± 0.012 for WT (n = 11) and 0.179 ± 0.005 (n = 11) for HET (−24%). Means were compared by unpaired, one-tailed t-test with Welch correction, p = 0.0007. Effect size, d = 1.75. (**B**) TARP γ -2,3,4,8 to PSD-95 ratio. Data were collected for 24 lanes from two blots containing 10 μg total PSD fraction per lane. Each blot contained six lanes WT and six lanes HET samples. Densities of all four TARPs were pooled. The mean ratio of TARPs to PSD-95 was 0.066 ± 0.003 (n = 12) for WT and 0.075 ± 0.003 (n = 12) for HET (+12%). Means were compared by unpaired, one-tailed t-test with equal variance, p = 0.017. Effect size, d = 0.93. (**C**) LRRTM2 to PSD-95 ratio. Data were collected for 36 lanes from three blots containing six WT and six HET samples, alternating 5 and 10 μg (3 each). The mean ratio of LRRTM2 to PSD-95 was 0.051 ± 0.003 (n = 17) for WT and 0.059 ± 0.003 (n = 17) for HET (+14%). Means were compared by paired, one-tailed t-test, p=0.0035. Effect size, d = 0.66. (**D**) Neuroligin-1 to PSD-95 ratio. Data were collected for 47 lanes from four blots two of which contained 5 μg and two 10 μg total PSD fraction per sample. Each blot contained 6 lanes WT and 6 lanes HET samples. The mean ratio of neuroligin-1 to PSD-95 was 0.114 ± 0.005 (n = 24) for WT and 0.115 ± 0.004 (n = 23) for HET (no difference). Means were compared by unpaired one-tailed t-test, p = 0.413. Effect size, d = 0.07. (**E**) Neuroligin-2 to PSD-95 ratio. Data were collected for 44 lanes from four blots containing 10 μg total PSD fraction per lane. Each blot contained six lanes WT and six lanes HET samples. The mean ratio of neuroligin-2 to PSD-95 was 0.071 ± 0.002 (n = 20) for WT and 0.078 ± 0.003 (n = 24) for HET (+9%). Means were compared by unpaired, one-tailed t-test with Welch correction, p=0.019. Effect size, d = 0.64. *p<0.005; **p<0.01; ***p<0.001

The following source data is available for figure 11:

**Source data 1.** Source data for ratios determined in *Figure 11* A through E.

of over-expression of PSD-95, suggesting that both manipulations increase the number of PDZ3 domains available for binding. Taken together, these results verify the prediction that a decrease in availability of synGAP in the PSD scaffold, increases the association of TARPs, LRRTMs, and neuroligin-2 with the PSD in vivo by releasing a restriction on binding to PDZ domains of PSD-95.

## Discussion

We have presented a new model for regulation of the composition of the spine postsynaptic membrane and PSD. It builds on previous findings that the PSD scaffold, in particular the PSD-95 complex, is dynamically regulated by activity (*Gray et al., 2006*; *Sturgill et al., 2009*), that activation of NMDARs and CaMKII leads to enhanced trapping of AMPARs within the PSD by binding to PDZ domains of PSD-95 (*Opazo et al., 2010*; *2012*), and that synGAP moves away from the PSD after phosphorylation by CaMKII (*Yang et al., 2013*; *Araki et al., 2015*). We propose that binding of the C-terminus of synGAP-α1 to the PDZ domains of PSD-95 restricts binding of TARPs, LRRTM2, neuroligin-2, and perhaps additional proteins. As a result, regulation of synGAP-α1's concentration and binding affinity for PDZ domains helps to determine the precise composition of the PSD at individual excitatory synapses (*Figure 12*). A corollary of this hypothesis is that synGAP plays an important role in limiting the size and strength of excitatory synapses by limiting and helping to regulate the available 'slots' that can bind AMPAR complexes (*Hayashi et al., 2000*; *Shi et al., 2001*; *Opazo et al., 2012*).We have provided support for this notion by demonstrating that phosphorylation of several sites in synGAP's regulatory domain by CaMKII or by PLK2 reduces the affinity of its PDZ ligand for all three of the PDZ domains of PSD-95. Both of these protein kinases are important regulators of synaptic strength (*Seeburg et al., 2008*; *Hell, 2014*). We have also verified a strong prediction of the hypothesis which is that PSDs from mice with a deletion of one copy of the synGAP gene will contain fewer copies of synGAP and more copies of other proteins that bind to PSD-95. Indeed, we have shown that PSD fractions from young *Syngap*[+/−] mice have ~24% less synGAP per molecule of PSD-95 than those from *wt* mice; and, in contrast, they have significantly more TARP proteins (~12%), LRRTM2 (~14%), and neuroligin-2 (~9%) per molecule of PSD-95. The ratio of neuroligin-1 to PSD-95 is not altered.

In the mouse brain, synGAP begins to be expressed at birth and its expression increases rapidly as synapses are forming during the first weeks after birth (*Vazquez et al., 2004*; *Barnett et al., 2006*). Newborns with complete deletion of *Syngap* appear normal, but die a few days after birth with movement defects and apparent seizure activity, indicating a defect as synapses are forming (*Kim et al., 2003*; *Vazquez et al., 2004*). *Syngap* heterozygous mice also appear normal and survive

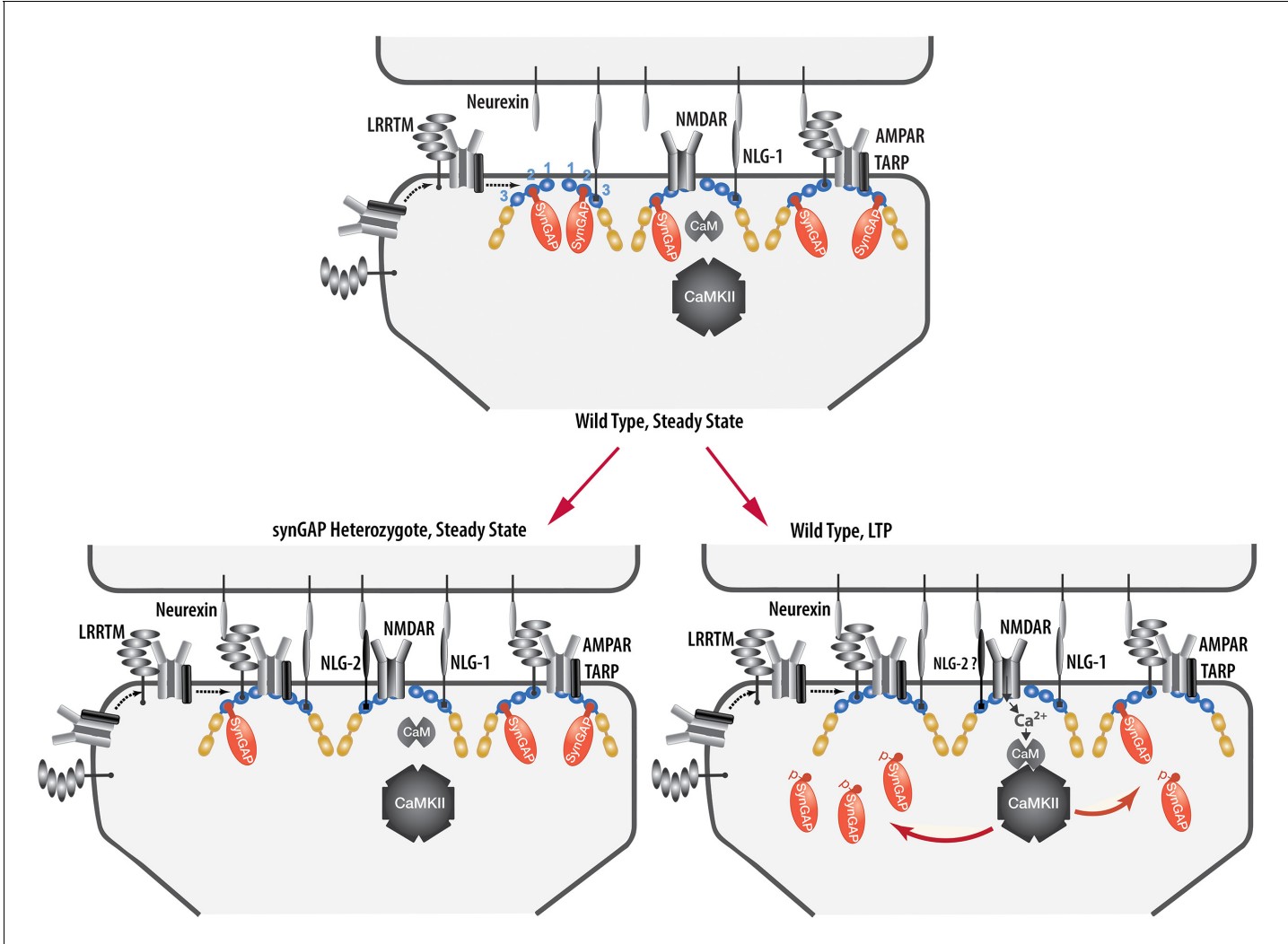

**Figure 12.** Cartoon model of rearrangement of PSD caused by synGAP haploinsufficiency or by phosphorylation of synGAP-α1 by CaMKII. Unphosphorylated synGAP-α1 (*Top*) binds to PDZ1, PDZ2 or PDZ3 of PSD-95, occupying as many as ~15% of its PDZ domains. The PDZ domains of PSD-95 are shown in blue and their numbers are indicated on the left pair of PSD-95 molecules. AMPARs that have been inserted into the extrasynaptic membrane by exocytosis associate with TARPs and with LRRTMs, both of which can bind to PDZ1 and PDZ2 of PSD-95. Neuroligin-1 (NLG-1) binds to PDZ3 of PSD-95. LRRTMs and NLG-1 also bind across the synaptic cleft to presynaptic neurexins. Induction of LTP (*Lower Right*) causes flux of calcium through NMDARs that activates CaMKII leading to phosphorylation of synGAP-α1 on sites in the regulatory domain. The affinity of synGAP-α1 for the PDZ domains decreases, allowing TARPs, LRRTMs, and perhaps NLG-2 to displace synGAP-α1 by binding to the PDZ domains. The shift in affinity of synGAP-α1 creates 'slots' that can be occupied by TARPs and LRRTMs bound to AMPARs, and by NLG-2, leading to strengthening of the synapse. SynGAP haploinsufficiency (*Lower Left*) results in a reduced amount of synGAP-α1 in the PSD. Similar to the model for LTP, this leaves more PDZ domains available to bind TARPs, LRRTMs, and NLG-2.

into adulthood with no apparent alterations in gross brain morphology, but with defects in behavior and synaptic plasticity (*Komiyama et al., 2002*). *Syngap*[+/-] neurons have higher average numbers of AMPARs at their synapses than *wt* (*Kim et al., 2003*; *Vazquez et al., 2004*), and, have, on average, larger spine heads (*Vazquez et al., 2004*; *Carlisle et al., 2008*). They form excitatory synapses precociously in culture (*Vazquez et al., 2004*) and also during postnatal cortical and hippocampal development (*Clement et al., 2012*), leading to an elevated E/I ratio (excitatory/inhibitory synaptic ratio) and derangement of critical developmental periods (*Clement et al., 2013*). While all of these influences could contribute to alterations in the composition of PSDs in *Syngap*[+/-] animals, several arguments suggest that the increased presence of TARPs, LRRTMs, and neuroligin-2 that we report in PSDs of young animals (*Figure 11*) is likely a primary, rather than a secondary effect of the

mutation. The effect of overexpression of synGAP-α1 in cultured forebrain neurons on development of spines (*Vazquez et al., 2004*) and on numbers of synaptic AMPARs (*McMahon et al., 2012*) is rapid, measurable within 12–24 hr of transfection and is not caused by mislocalization of the exogenous synGAP- α1. These effects are predicted by our model and do not depend on the in vivo context of the neurons. They are also not dependent on the GAP activity of synGAP and they require the PDZ ligand. While it is possible that a presently unknown direct effect on protein expression produces these changes, that effect would have to be specific for synGAP-α1 and require its PDZ ligand. No evidence has emerged for a direct effect of the PDZ ligand on protein expression. Thus, the hypothesis we have put forward is the most parsimonious explanation for the specific effects of synGAP-α1 on PSD composition. Similarly, the change in protein composition of PSDs in *Syngap$^{+/-}$* animals (*Figure 11*) cannot be explained by a two-fold reduction in RasGAP and RapGAP activity. SynGAP produces a much larger fold increase in GTPase activity of Rap than of Ras (*Walkup et al., 2015* and references therein). Therefore, loss of half of the GAP activity would be predicted to increase the level of active Rap even more than that of active Ras. Thus, one would expect a relative increase in endocytosis of AMPARs compared to exocytosis (*Zhu et al., 2002*). That would result in fewer AMPAR binding proteins in the PSD; not more, as we observe. For now, application of Occam's razor strongly favors the hypothesis we have presented. Future experiments will be necessary to determine the quantitative significance of synGAP-α1's restriction of PDZ domain binding sites at various times during development and during acute stimulation of synapses.

Because binding between molecules is driven by their concentrations and also by the inherent affinity between the binding components, we predict that phosphorylation of synGAP-α1 by CaMKII or PLK2 can alleviate the restriction, enabling reconfiguration of the PSD scaffold. Thus, acute phosphorylation of synGAP-α1 by CaMKII following activation of NMDARs during induction of LTP could initiate rearrangement of the composition of PSDs of individual synapses by causing an increase in equilibrium binding in the PSD of AMPARs associated with TARPs and LRRTMs, and of other PSD-95 binding proteins, as they diffuse from perisynaptic locations. It will be important to test this prediction in future experiments.

Dephosphorylation of synGAP-α1 after its movement away from PDZ domains might be expected to allow synGAP-α1 to displace the newly added TARPs and LRRTMs and reverse the addition of new AMPARs. Thus, if this model is correct, additional processes occurring later in the consolidation of LTP would be needed to stabilize the newly added AMPARs in the synapse and/or to permanently decrease the number of synGAP-α1 molcules per PSD-95 in potentiated synapses. Such processes could include degradation of the phosphorylated synGAP-α1 and its replacement by newly synthesized alternatively-spliced isoforms that lack the C-terminal PDZ binding domain (*Li et al., 2001*; *McMahon et al., 2012*).

Several studies have shown that an increase in localization of neuroligin-2 at excitatory synapses is mediated by binding to PSD-95 and increases the ratio of excitatory to inhibitory synaptic contacts (*Prange et al., 2004*; *Levinson et al., 2005*; *2010*). It seems likely that the restriction by synGAP-α1 of binding of neuroligin-2 to PDZ3 of PSD-95 would be more significant during development, during formation of new synapses, or perhaps in later phases of consolidation of LTP in adults. It is not clear whether a pool of perisynaptic neuroligin-2 exists at adult synapses that could be recruited to new synaptic sites over a few minutes after phosphorylation of synGAP-α1. Nonetheless, the small but significant increase in neuroligin-2 that we observe in excitatory PSDs from *Syngap$^{+/-}$* mice, as well as the increased steady state amounts of TARPs and LRRTMs would increase the overall excitatory/inihibitory (E/I) balance of synapses onto neurons in the mutant mice, and perhaps also in humans with *SYNGAP1* haploinsufficiency.

We note that the reduction in affinity of synGAP-α1 for PDZ domains of PSD-95 after phosphorylation by CaMKII is apparently not sufficient for complete dispersal of synGAP-α1 away from the PSD, although it is likely necessary. Mutated synGAP-α1 missing the PDZ ligand (synGAPΔSXV) cannot bind to PDZ domains, yet still localizes to synaptic spines (*Vazquez et al., 2004*). Furthermore, *Yang et al. (2013)* showed by immunoelectron microscopy that both α1 and α2 synGAP isoforms localize to the PSD core, and *McMahon et al. (2012)* showed by mass spectrometry that α2 isoforms are present in isolated PSDs. These data mean that reduction of affinity of synGAP for PDZ domains may be sufficient to decrease its ability to compete for binding slots on PSD-95; however, complete detachment of synGAP from the PSD in vivo likely requires additional events.

The functional significance of our finding that r-synGAP-α1 contains a high affinity binding site for $Ca^{2+}$/CaM is less clear. We have shown that binding of $Ca^{2+}$/CaM alters the conformation of the carboxyl terminal regulatory domain of r-synGAP-α1 allowing CDK5 to phosphorylate additional sites; the binding also reduces the affinity of r-synGAP-α1 for PDZ3 by ~25%. However, the consequences of these two effects for synaptic function are not known. Once again, the high copy number of synGAP in the PSD may provide a clue. The biochemical events initiated by $Ca^{2+}$ flux through NMDARs that lead to changes in synaptic strength (*Sjostrom and Nelson, 2002*) are initiated by formation of transient and limiting concentrations of $Ca^{2+}$/CaM in the spine (*Markram et al., 1998*; *Pepke et al., 2010*). Approximately ten regulatory enzymes compete for binding of, and activation by, this $Ca^{2+}$/CaM (*Kennedy, 2013*). Because of the abundance of synGAP in the PSD, the high affinity binding site for $Ca^{2+}$/CaM on synGAP will compete effectively for the newly formed $Ca^{2+}$/CaM and may act as a $Ca^{2+}$/CaM buffer.

Haploinsufficiency of *SYNGAP1* in humans is the cause of ~2–9% of cases of nonsyndromic cognitive disability with co-morbid Autism Spectrum Disorder or Epilepsy (*Berryer et al., 2013*). The reduced amount of synGAP and resulting decrease in its ability to compete for PDZ domains are likely as significant as the reduction in synaptic Ras/Rap GAP activity in the pathology of these disorders. An increase in the E/I balance of synapses onto neurons in the forebrain of affected individuals is predicted by our results with *Syngap*$^{+/-}$ mice, and could be responsible for the symptoms of cognitive disability, ASD, and/or epilepsy. It would be worth considering whether pharmaceutical agents could be designed that would bind to PDZ domains of PSD-95 (e.g. *Cui et al., 2007*) and compensate for the reduced level of synGAP.

## Materials and methods

### Cloning, expression and purification of r-synGAP

Soluble, recombinant synGAP-α1 (r-synGAP-α1), comprising residues 103–1293 in synGAP A1-α1 (118–1308 in synGAP A2-α1), or sr-synGAP, comprising residues 103–725 in synGAP A1-α1 (118–740 in synGAP A2-α1), was purified from *E. coli* as previously described (*Walkup et al., 2015*). The isoform names and residue numbering are taken from ref. (*Walkup et al., 2015*). Henceforth, except where indicated, we use residue numbering corresponding to synGAP A1-α1.

Briefly, a pET-47b(+) plasmid (EMD Millipore, Temecula, CA; catalog no. 71461) containing r-synGAP-α1 cDNA (AF048976) fused to an N-terminal 6x Histidine Tag and a PreScission Protease cleavage site was transformed into the Rosetta2(DE3) *E. coli* strain (EMD Millipore, catalog no. 71397) for protein expression. Bacterial pellets were harvested by centrifugation and lysed by microfluidization in a ML-110 microfluidizer (Microfluidics). Soluble r-synGAP-α1 was purified on Talon Metal Affinity Resin (Clontech, Mountain View, CA; catalog no. 635503), and concentrated by ultrafiltration through a 30 kDa cutoff-filter (Thermo Scientific, Waltham, MA; catalog no. 88531) for r-synGAP-α1 or 9 kDa cutoff-filter (Thermo Scientific, catalog no. 89885A) for sr-synGAP. Concentrated samples of r-synGAP-α1 were exchanged into storage buffer (20 mM Tris, pH 7.0; 500 mM NaCl, 10 mM TCEP, 5 mM $MgCl_2$, 1 mM PMSF, 0.2% Tergitol Type NP-40, and Complete EDTA-free protease inhibitor) by ultrafiltration, flash-frozen in liquid nitrogen, and stored at -80°C. Sr-synGAP was further purified on a size exclusion column prior to storage (*Walkup et al., 2015*).

### Cloning, expression and purification of PDZ domains from PSD-95

Soluble recombinant PDZ domains, comprising residues 61–151 (PDZ1), 155–249 (PDZ2), 302–402 (PDZ3), 61–249 (PDZ12), and 61–403 (PDZ123) from murine PSD-95 (Q62108) were purified from *E. coli* as previously described (*Walkup and Kennedy, 2014*, *2015*) with the modifications below.

Briefly, pJExpress414 plasmids (DNA2.0, catalog no. pJ414) containing codon optimized PDZ domains were transformed into the BL21(DE3) *E. coli* strain (EMD Millipore, catalog no. 70235–3) for protein expression. Single colonies of BL21(DE3) cells harboring pJExpress414 plasmids were grown overnight at 37°C in lysogeny broth (LB) (Teknova, Hollister, CA; catalog no. L9110) supplemented with 100 μg/ml carbenicillin. Overnight cultures were diluted 1:500 into LB medium and grown at 37°C until cultures reached an O.D.$_{600}$ of 1.0. IPTG was added to a final concentration of 0.2 mM, and cultures were grown for an additional 4.5 hr at 37°C. Bacterial pellets were harvested by centrifugation and lysed using non-ionic detergent (BugBuster, EMD Millipore) and Ready-Lyse (Epicentre,

Madison, WI). Soluble PDZ1, PDZ2 and PDZ12 domains were purified on GluN2B peptide (GAGS-SIESDV) PDZ Ligand Affinity Resin (*Walkup and Kennedy, 2014*) by eluting with 400 µg/ml SIETEV peptide. PDZ3 and PDZ123 were purified on CRIPT peptide (GAGNYKQTSV) PDZ Ligand Affinity Resin (*Walkup and Kennedy, 2014*) by eluting with 400 µg/ml YKQTSV peptide. PDZ domains were concentrated by ultrafiltration through a 3 kDa Amicon Ultracentrifugal Filter Unit (EMD Millipore, catalog no. UFC 900396). The PDZ peptide ligands were removed from PDZ domains by dialysis into storage buffer (50 mM HEPES, pH 7.5; 100 mM NaCl, 5 mM TCEP, 1 mM PMSF, and Complete EDTA-free protease inhibitor). Purified PDZ domains (>99% pure; 45–610 µM; *Figure 1*) were flash-frozen in liquid nitrogen, and stored at -80°C.

## SDS-PAGE, immunoblotting and assessment of protein purity and yield

We used SDS-PAGE to determine purity of proteins and to quantify binding to PDZ domain resin. Protein samples were diluted 1:3 into 3x Laemmli buffer (100 mM Tris HCl, pH 6.8; 2.1% SDS, 26% glycerol, 7.5% β-mercaptoethanol, and 0.01% bromophenol blue) and heated to 95°C for 3 min before fractionation on 8% SDS-PAGE gels at 165 V in 25 mM Tris base, 192 mM glycine, 0.1% SDS. Proteins were stained with Gel Code Blue (Thermo Scientific, catalog no. 24592), imaged on a Li-Cor Odyssey Classic Infrared Imaging System (Li-Cor Biosciences, Lincoln, NE) at 700 nm, and quantified with Licor Image Studio Software (v4.0.21) against standard curves of BSA (catalog no. A7517-1VL) and lysozyme (catalog no. L4631-1VL) purchased from Sigma-Aldrich, St. Louis, MO. The protein standards were loaded onto each gel in lanes adjacent to the protein samples. Molecular weights of stained proteins were verified by comparison to Precision Plus Protein All Blue Standards (BioRad, Irvine, CA; catalog no. 161–0373).

For immunoblotting, proteins fractionated by SDS-PAGE were electrically transferred to low fluorescence PVDF membranes (Thermo Scientific, catalog no. 22860) in 25 mM Tris, 200 mM glycine, and 20% methanol. Membranes were washed with 50 mM Tris-HCl, pH 7.6; 150 mM NaCl (TBS) followed by blocking with Odyssey Blocking Buffer (Li-Cor Biosciences, catalog no. 927–50000). Membranes were washed in TBS supplemented with 0.1% Tween 20 (TBS-T) before incubation in Odyssey Blocking Buffer containing 1:1000 diluted rabbit anti-synGAP (ThermoFisher Pierce, Waltham, MA catalog no. PA1-046) or 1:1500 BSA-free anti-TetraHis (Qiagen, Hilden, Germany, catalog no. 34670). Bound antibodies were detected with 1:10,000 goat anti-mouse Alexa-Fluor 680 (Life Technologies, catalog no. A-21057) or 1:10,000 goat anti-rabbit Alexa-Fluor 680 (Life Technologies, Carlsbad, CA; catalog no. A-21109) visualized with a Li-Cor Odyssey Classic Infrared Imaging System and quantified with Li-Cor Image Studio Software.

## Synthesis of PDZ domain affinity resins

PDZ domain affinity resins (PDZ1, residues 61–151; PDZ2, residues 155–249; PDZ3, residues 302–402; PDZ12, residues 61–249; PDZ123, residues 61–402 from murine PSD-95) were prepared by the HaloTag-HaloLink method as previously described (*Walkup and Kennedy, 2014*, *2015*). Briefly, bacterial cell pellets containing PDZ domain-HaloTag fusion proteins were resuspended in 10 ml/g of Purification Buffer, and lysed by three passes through a ML-110 microfluidizer. The lysate was clarified by centrifugation, added to HaloLink resin (Promega, Madison, WI; catalog no. G1915), and mixed with continuous agitation for 1.5 hr at 4°C on an end-over-end mixer. Unbound protein was separated from the resin by centrifugation and the PDZ-HaloTag-HaloLink resin was resuspended, transferred to a column, and allowed to settle. The resin was extensively washed and then stored at 4°C in a buffer supplemented with 0.05% NaN₃. The resin was used or discarded within 1 week of preparation. The densities of PDZ domains on the resin varied from 50 to 100 pmol of PDZ123 domain and from 200 to 500 pmol of PDZ1, PDZ2, PDZ3, or PDZ12 domains per µl resin.

## Assay for binding of r-synGAP-α1 to PDZ domain resin

Phosphorylated or nonphosphorylated r-synGAP-α1 (500 nM, 200 µl) was mixed with 20 µl of Affinity Resin containing PDZ1, PDZ2, PDZ3, PDZ12, or PDZ123 domains, pre-equilibrated with Binding/Wash Buffer (25 mM Tris, pH 7.0; 150 mM NaCl, 1 mM MgCl₂, 0.5 mM TCEP, 0.2% Tergitol, 0.5 mM EDTA) in a cellulose acetate spin cup (ThermoFisher Pierce, catalog no. 69702) for 60 min on an end-over-end mixer. In some experiments, 2.5 µM CaM and 0.5 mM CaCl₂ were included to test the effect of binding of $Ca^{2+}$/CaM to r-synGAP-α1 on synGAP's affinity for PDZ domains. After

the incubation, the resin in the spin cup was centrifuged for 2 min at 1500 x g to remove unbound protein, and the resin was washed 4 times with 200 µl of Binding/Wash Buffer. To elute bound protein, 100 µl of 1x Laemmli Buffer was added and the resin was incubated for 5 min at room temperature. The eluted protein was collected by centrifugation at 6000 x g for 2 min, fractionated by SDS-PAGE, stained with Gel Code Blue, and quantified on a Li-Cor Classic as described above. Integrated intensities reflecting the amount of bound r-synGAP-α1 were determined with Li-Cor software and plotted with Prism (v6.0d, GraphPad Software, La Jolla CA). There is no detectable non-specific binding of r-synGAP-α1 to unsubstituted resin or of proteins lacking PDZ domain ligands to the affinity resins (*Walkup and Kennedy, 2014*).

## Phosphorylation of r-synGAP-α1 by CaMKII, CDK5 and PLK2 for PDZ binding assays

Phosphorylation of r-synGAP-α1 by CaMKII, CDK5, and PLK2 was carried out immediately prior to PDZ binding assays, as previously described (*Walkup et al., 2015*). Reaction mixtures contained 50 mM Tris-HCl, pH 8.0; 10 mM $MgCl_2$, 0 or 0.7 mM $CaCl_2$, 0.4 mM EGTA, 30 µM ATP, 0 or 3.38 µM calmodulin, 10 mM DTT, 725 nM r-synGAP-α1. Reactions contained no kinase, 10 nM CaMKII, 230 nM CDK5/p35 (EMD Millipore, catalog no. 14-477M) or 230 nM PLK2 (Life Technologies, catalog no. PV4204). Mixtures for CDK5 and PLK2 did not contain $CaCl_2$ or CaM. Samples were quenched by addition of 1/3 volume of 50 mM Tris, pH 8.0; 0.4 M NaCl, 10 mM $MgCl_2$, 0.8% tergitol (Type NP-40), 6 µM autocamtide-2 related inhibitory peptide (Genscript, Piscataway, NJ; catalog no. RP10271), 90 µM roscovitine (Sigma-Aldrich) and 40 mM EGTA at the indicated times. When we planned to add $Ca^{2+}$/CaM during the subsequent incubation with resin, the EGTA was omitted. Samples were stored on ice until their use in PDZ domain binding assays.

## Stoichiometry and rate of phosphorylation of r-synGAP-α1 and histone H1 by CaMKII and/or CDK5

Phosphorylation of 725 nM r-synGAP-α1 by 10 nM CaMKII was carried out as previously described (*Walkup et al., 2015*) in reaction mixtures containing 50 mM Tris-HCl, pH 8.0, 10 mM $MgCl_2$, 0 or 0.7 mM $CaCl_2$, 0.4 mM EGTA, 500 µM $[\gamma\text{-}^{32}P]$-ATP (375 cpm/pmol) (6000 Ci/mmol, Perkin Elmer, Waltham, MA; catalog no. BLU002Z/NEG002Z), 0 or 3.4 µM calmodulin, 10 mM DTT. Phosphorylation of 286 nM r-synGAP-α1 and 4.3 µM Histone H1 (New England Biolabs, Ipswich, MA; catalog no. M2501S), by CDK5/p35 (EMD Millipore, catalog no. 14-477M) or CDK5/p25 (EMD Millipore, catalog no. 14–516) was carried out in the same reaction mixture containing 110 nM CDK5/p35 or CDK5/p25 but no CaMKII. After fractionation by SDS-PAGE, phosphorylated proteins were quantified with a Typhoon LA 9000 phosphorimager (GE Healthcare Life Sciences, Pittsburgh, PA) as previously described (*Walkup et al., 2015*). Relative densities were converted to pmol phosphate by comparison to densities of standard amounts of $[\gamma\text{-}^{32}P]$-ATP. The stoichiometry of phosphorylation was calculated by dividing mol of incorporated phosphate by mol of r-synGAP-α1 loaded per lane.

## Measurement of affinity of r-synGAP-α1 for PDZ domains by surface plasmon resonance (SPR)

We used a 'competition in solution' method (also called 'affinity in solution') (*Nieba et al., 1996*; *Lazar et al., 2006*; *Abdiche et al., 2008*) to measure the affinity of r-synGAP-α1 for PDZ domains. In this method, PDZ domains are immobilized on the chip surface and used to capture and measure the concentration of free r-synGAP-α1 in pre-equilibrated mixtures of a constant amount of r-synGAP-α1 with varying amounts of soluble recombinant PDZ domains. Experiments were performed on a Biacore T200 (GE Healthcare Life Sciences). Purified PDZ domains (PDZ1, PDZ2, PDZ3, PDZ12, PDZ123 from PSD-95) were coupled to Series S CM5 Sensor Chips (GE Healthcare Life Sciences, catalog no. BR-1005-30) by the amine coupling protocol specified in the Biacore T200 Control Software with reagents purchased from GE Healthcare Life Sciences. Sensor surfaces were activated by applying a 1:1 mixture of 50 mM N-hydroxysuccinimide (NHS): 200 mM 1-ethyl-3-(3-dimethylaminopropyl) carbodiimide hydrochloride (EDC) provided in the Biacore Amine Coupling Kit (GE Healthcare Life Sciences, catalog no. BR-1000-50) dissolved in HBS-N running buffer (degassed 0.01 M HEPES pH 7.4, 0.15 M NaCl) (GE Healthcare Life Sciences, catalog no. BR-1006-70). PDZ domains were diluted to 0.1–5 µM in Biacore Sodium Acetate Buffer [10 mM sodium acetate, pH 4.0 (GE Healthcare Life

Sciences, catalog no. BR-1003-49) for PDZ1 and PDZ3; pH 4.5 for PDZ2; pH 5 for PDZ12 and PDZ123]. PDZ domains were injected into flow cells 2 and 4 until 200 to 400 RU (resonance units) of PDZ domain were immobilized. Flow cells 1 and 3 were left blank to be used as reference surfaces. Ethanolamine (1 M, pH 8.5) was injected for 7 min at 10 μl/min to block remaining active sites on all four flow cells.

A calibration curve was prepared by applying samples of 0 to 50 nM r-synGAP-α1 prepared by two-fold serial dilution of 50 nM r-synGAP-α1 into 1x HBS-EP+ buffer (degassed 0.01 M HEPES, pH 7.4; 0.15 M NaCl, 3 mM EDTA, 0.005% v/v Surfactant P20; GE Healthcare Life Sciences, catalog no. BR-1006-69) to the chip and recording the maximum RU for each concentration. Samples for calibration were incubated for 2 hr at room temperature before randomized injection onto the chip surface at 25°C at 10 μl/min for 200 s over all four flow cells. Between each sample injection, the chip was regenerated by injecting 1 M $MgCl_2$ at 100 μl/min for 60 s, waiting 180 s for the baseline to stabilize, then injecting a second pulse of $MgCl_2$ solution, waiting 300 s for the baseline to stabilize, and finally executing a 'carry over control injection' in which HBS-EP+ buffer is flowed over the chip surface at 40 μl/min for 30 s. Mixtures of r-synGAP-α1 and PDZ domains were prepared by 1:1 dilution of 50 nM r-synGAP-α1 with two-fold serial dilutions of PDZ domains (0–20 μM PDZ1, PDZ2, PDZ3, PDZ12 or 0–160 μM PDZ123) in HBS-EP+ buffer to produce mixtures containing 25 nM r-synGAP-α1 and 0–10 μM PDZ1, PDZ2, PDZ3, PDZ12 or 0–80 μM PDZ123. For each mixture of r-synGAP-α1 and PDZ domain, the different concentrations were injected randomly and a series of sensorgrams were recorded as described for the calibration curve.

Sensorgrams were processed with Biacore T200 Evaluation Software, (ver. 3.0, GE Healthcare Life Sciences). The y-axes were zeroed at the baseline for each cycle and x-axes were aligned at the injection start. Bulk refractive index changes and systematic deviations in sensorgrams were removed by subtracting the responses in reference flow cells corresponding to the sample flow cells (e.g. 2–1, 4–3). The averaged sensorgrams for 0 nM r-synGAP-α1 were then subtracted from sensorgrams for all other concentrations. The concentrations of free r-synGAP-α1 in each mixture with PDZ domains was determined from the calibration curve, exported into Prism, and plotted against the log of PDZ domain concentration. The curve was fit to the equation:

$$synGAP_{free} = \frac{(synGAP_{tot} - PDZ_{tot} - K_D)}{2} \pm \sqrt{\frac{(PDZ_{tot} + synGAP_{tot} + K_D)^2}{4} - (PDZ_{tot} \times synGAP_{tot})}$$

This equation (from the Biacore T200 Software Handbook) assumes the existence of a single binding site between PDZ domain and r-synGAP-α1. Equilibrium dissociation constants ($K_D$) for binding to PDZ1, PDZ2, and PDZ3 were determined from the best fit curves as described in the Biacore T200 Software Handbook (p. 210). The experimental curves deviated slightly from the equation at higher concentrations of soluble PDZ domain because of a low affinity association between PDZ domains bound to the chip and soluble free PDZ domains. The deviation is most obvious for phosphorylated r-synGAP-α1 which has a relatively low affinity for PDZ123. We determined that these deviations had only a small effect (<~5%) on the calculated $K_D$'s by comparing $K_D$'s after fitting the curves including or excluding points for high concentrations of PDZs at which the r-synGAP-α1 concentration appeared to plateau above zero. PDZ12 and PDZ123 contain more than a single PDZ domain binding site. It is not possible to derive a unique equation incorporating two or three binding sites when the affinities of the multiple sites are similar. Therefore, single 'apparent' equilibrium dissociation constants ($K_{Dapp}$) were determined by obtaining the best fit of the data with the equation for a single binding site.

## Binding of r-synGAP-α1 to CaM-Sepharose

Rosetta2(DE3) cells containing sr-synGAP or r-synGAP-α1 were lysed in Lysis buffer as described in *Walkup et al. (2015)* except that the buffer contained 200 mM NaCl, 0 or 5 mM $CaCl_2$, and 0 or 10 mM EGTA. The resuspended cells were lysed by sonication with a Digital Sonifier 450 Cell Disruptor (Branson, Wilmington, NC) for two passes at 90 s/pass (15% power, 1.0 s on, 1.5 s off), and insoluble material was removed by centrifugation at 16,000 × g for 40 min at 4°C. Clarified cell lysate (1.7 ml) containing sr-synGAP or r-synGAP-α1 (~6 mg/ml total protein) was incubated with end-over-end mixing for 60 min with 0.3 ml CaM-Sepharose 4B (GE Healthcare Life Sciences, catalog no. 17-0529-01) or control Sepharose 4B (GE Healthcare Life Sciences, catalog no. 17-0120-01). The resin

was pipetted into a BioSpin column (Bio-Rad, catalog no. 732–6008) and washed with 12 ml (40 column volumes) of Lysis/Wash Buffer. Bound protein was eluted with 1.2 ml (4 column volumes) of Lysis/Wash Buffer containing 100 mM EGTA. Eluted proteins (30 μl aliquot) were resolved by SDS-PAGE and transferred to a PVDF membrane which was probed with anti-synGAP or BSA-free anti-TetraHis as described above.

## Measurement of affinity of CaM for r-synGAP-α1 by SPR

Direct binding of r-synGAP-α1 to $Ca^{2+}$/CaM immobilized on a chip was assayed on a Biacore T200 with a Series S Sensor Chip CM5. CaM (Enzo Life Sciences, Farmingdale, NY; catalog no. BML-SE325-0001) was coupled to the chip by the amine coupling protocol specified in the Biacore T200 Control Software, as described above. Purified, lyophilized CaM (250 μg) was resuspended in water and exchanged into Biacore Sodium Acetate, pH 4.0 Buffer (10 mM sodium acetate, pH 4.0) with an Amicon Ultra-0.5 ml centrifugal filter with a 3 kDa molecular weight cutoff (EMD Millipore, catalog no. UFC500396). CaM was further diluted to 0.5 nM in 10 mM Sodium Acetate, pH 4.0, and injected into flow cells 2 and 4 until 50 RU of CaM were immobilized (~7 min at a flow rate of 10 μl/min). Flow cells 1 and 3 were left blank to be used as reference surfaces. Ethanolamine (1 M, pH 8.5) was injected for 7 min at 10 μl/min to block remaining active sites on all four flow cells. R-synGAP-α1 (0 nM to 75 nM) in 1x HBS-EP+ running buffer supplemented with 10 mM $CaCl_2$, was injected in triplicate at 25°C at 100 μl/min for 75 s over all four flow cells. Different concentrations of r-synGAP-α1 were applied in randomized order. After injection ended, dissociation was monitored in each flow cell for 500 s. Regeneration of the sensor chip was performed by injecting 50 mM NaOH (GE Healthcare Life Sciences, catalog no. BR-1003-58) at 100 μl/min for 30 s, waiting 180 s for the baseline to stabilize, then injecting a second pulse of NaOH, waiting 240 s for the baseline to stabilize, and finally executing a 'carry over control injection.' Sensorgrams were processed using the Biacore T200 Evaluation Software, version 3.0, as described above. Resonance units of bound r-synGAP-α1 at equilibrium were exported into Prism and plotted against the concentrations of r-synGAP-α1. The data were fit globally to a hyperbolic curve by nonlinear regression to determine equilibrium dissociation constants ($K_D$).

## Determination of affinity of $Ca^{2+}$/CaM for sr-synGAP and r-synGAP-α1 by equilibrium binding to CaM-Sepharose

Purified sr-synGAP and r-synGAP-α1 were diluted to 1 to 500 nM in Binding/Wash Buffer (50 mM Tris, pH 7.5; 200 mM NaCl, 5 mM TCEP, 2 mM $CaCl_2$). Aliquots of diluted r-synGAP-α1 (300 μl) were incubated with end-over-end mixing for 60 min with 50 μl of CaM-Sepharose 4B in a screw cap spin column (Thermo Scientific, catalog no. 69705). Concentrations of sr-synGAP and r-synGAP-α1 were 20–3,000 fold below the ligand binding capacity of the CaM-Sepharose resin. Unbound protein was removed by centrifugation at 4000 x g for 30 s. Beads were washed in Binding/Wash Buffer (250 μl, 5 volumes), and bound protein was eluted with 50 μl of 1x Laemmli buffer with 10% β-mercaptoethanol. Eluted proteins were fractionated by SDS-PAGE and transferred to a PVDF membrane as described above. Blots were probed with 1:1000 anti-synGAP or 1:1500 BSA-free anti-TetraHis antibodies and quantified on a Li-Cor Classic as described above. Integrated intensities reflecting the amount of bound r-synGAP-α1 were determined with Li-Cor software and plotted against the corresponding concentrations of r-synGAP-α1 with Prism software. The data were fit to a hyperbolic curve by nonlinear regression to determine the dissociation constant ($K_D$).

## Preparation of PSD fractions

*Syngap*[+/-] mice were bred from a knockout strain created in our lab (*Vazquez et al., 2004*). The mutant strain has been deposited in the Mutant Mouse Resource & Research Center at University of California, Davis, listed as MMRRC:037374-UCD. All procedures were approved by the Caltech Institute Animal Care and Use Committee. PSD fractions were prepared as previously described (*Cho et al., 1992*) from six *wt* and six *Syngap*[+/−] mouse litter mates matched by age (7–12 weeks), and sex (*wt*, 1 female, 5 males; *Syngap*[+/−], 2 female, 4 male). The mice were killed by cervical dislocation and forebrains were dissected and rinsed in Buffer A (0.32 M sucrose, 1 mM NaHCO3, 1 mM MgCl2, 0.5 mM CaCl2, 0.1 mM PMSF, 1 mg/l leupeptin). Forebrains from each set of six mice were pooled and homogenized with 12 up and down strokes at 900 rpm in 14 ml Buffer A. Homogenates

were diluted to 35 ml in Buffer A and centrifuged at 1400 × g for 10 min. The pellet was resuspended in 35 ml Buffer A, homogenized (3 strokes), and centrifuged at 710 g for 10 min. Supernatants from the two centrifugations were combined and centrifuged at 13,800 g for 10 min. The pellet was resuspended in 8 ml of Buffer B (0.32 M sucrose, 1 mM $NaH_2CO_3$), homogenized with 6 strokes and layered onto a sucrose gradient (10 ml each of 0.85 M, 1.0 M, and 1.2 M sucrose in 1 mM $NaH_2CO_3$ buffer). The gradient was centrifuged for 2 hr at 82,500 g in a swinging bucket rotor. The synaptosome-enriched layer at the interface of 1.0 and 1.2 M sucrose was collected, diluted to 15 ml with Solution B and added to an equal volume of Buffer B containing 1% Triton. The mixture was stirred for 15 min at 4°C and centrifuged for 45 min at 36,800 g. The pellet containing the PSD-enriched, Triton-insoluble fraction was resuspended in 800–1000 µl of 40 mM Tris pH 8 with a 21 gauge needle and 1 ml syringe, and further solubilized by hand in a teflon glass homogenizer. Samples were aliquoted and stored at −80°C.

## Quantification of proteins in PSD fractions

Equal amounts of protein from each PSD sample (5–15 µg) were dissolved in SDS-PAGE sample buffer (33 mM Tris HCl, pH 6.8; 0.7% SDS, 10% glycerol, 2.5% β-mercaptoethanol, and 0.003% bromophenol blue), heated at 90°C for 5 min, fractionated on polyacrylamide gels (8% or 10%), and electrically transferred to PVDF membranes in 25 mM Tris, 150 mM glycine, 2% methanol at 250V for 2.5 hr at 4°C, as described above. Membranes were blocked with Odyssey blocking buffer (Licor Biosciences) and then incubated in a primary antibody solution of 5% BSA in TBS-T overnight at 4C, as described above. Primary antibodies included mouse-anti-PSD-95 (ThermoFisher, catalog no. MA1-046 [clone 7E3-1B8], RRID AB_2092361, dilution 1:10,000), rabbit-anti-SynGAP (Pierce, catalog no. PA1-046, RRID AB_2287112, dilution 1:3500), rabbit-anti-TARP (ɣ-2,3,4, and 8; EDM Millipore, catalog no. Ab9876, RRID AB_877307, dilution 1:300), rabbit-anti-LRRTM2 (Pierce, catalog no. PA521097, RRID AB_11153649, dilution 1:3000), mouse-anti-neuroligin-1 (Sigma, catalog no. sab5201464, RRID AB_2570548, dilution 1:250), and rabbit-anti-neuroligin-2 (Synaptic Systems, Gottingen, Germany, catalog no. 129202, RRID AB_993011, dilution 1:1000). The membranes were then washed 3-times in TBS-T. The membrane was incubated with secondary antibodies (Alexa Fluor 680-goat-anti-mouse IgG (Life Technologies, catalog no. A21057; 1:10,000) or IRDye800-goat-anti-rabbit IgG (Rockland, Limerick, PA; catalog no. 611-132-122; 1:10,000) for 45 min at room temperature in 5% nonfat milk in TBS-T, then washed 3 times in TBS-T, then twice in TBS prior to scanning. For most experiments, each blot contained 6 duplicate samples of PSD fractions from *wt* and the same number from S*yngap*$^{+/-}$ mice. Each blot was incubated with a mixture of two primary antibodies; mouse-anti-PSD-95 and the antibody against neuroligin-1, neuroligin-2, TARPs, LRRTM2, or synGAP. Then the blots were incubated with a mixture of the appropriate secondary antibodies. For measurement of neuroligin-1, both PSD-95 and neuroligin-1 were detected by the same goat anti-mouse secondary; the bands were physically separated on the gel and were quantified independently. Bound antibodies were visualized in the appropriate fluorescent channels with an Odyssey Classic Infared Imaging System (Li-Cor Biosciences).

Before running samples for quantification, we determined the amount of PSD sample that would result in signals for each protein that were strong enough for measurement and not saturated. To quantify the densities of the bands, each visual image was first set to high brightness in order to capture the boundaries of the signals for each band. The images were then used as a template in Li-Cor software to draw rectangular regions of interest around protein bands, and around identically sized background regions in the same lane. Background densities were subtracted from each protein signal. The digital data read by the Li-Cor software is unchanged by the visualization settings and is linear over several orders of magnitude. For each lane, the ratio of the integrated density of each of the five proteins to the integrated density of PSD-95 was calculated. For three gels, one outlier measurement (defined as greater than 2 times the standard deviation of the mean [S.E.]) was excluded from the calculation. The mean and S.E of the ratios were determined for *wt* and *Syngap*$^{+/-}$ PSD fractions. The means were compared by t-tests performed with Prism software as indicated in the legend of *Figure 11*. The effect size was measured by Cohen's d (ratio of the difference in the mean to the pooled standard deviations of the measurements). Cohen's d between 0.5 and 0.8 is generally considered a 'medium' effect size, greater than 0.8 is generally considered a large effect size. For four of the proteins, the number of measurements was sufficient to determine a significant difference

between *wt* and *Syngap$^{+/-}$*. In the case of neuroligin-1, the means were identical after 24 individual measurements of each sample.

## Acknowledgements

This work was supported in part by grants from the Gordon and Betty Moore Foundation (Center for Integrative Study of Cell Regulation), the Hicks Foundation for Alzheimer's Research, the Allen and Lenabelle Davis Foundation, and from National Institutes of Health Grant MH095095 to MBK. WGW IV was supported by the National Science Foundation Graduate Research Fellowship under Grant No. 2006019582, and the National Institutes of Health under Grant No. NIH/NRSA 5 T32 GM07616. The Protein Expression Center is supported by the Beckman Institute.

## Additional information

### Competing interests

MBK: Reviewing editor, *eLife*. The other authors declare that no competing interests exist.

### Funding

| Funder | Grant reference number | Author |
| --- | --- | --- |
| National Science Foundation | 2006019582 | Ward G Walkup IV |
| National Institutes of Health | NRSA T32 GM07616 | Ward G Walkup IV |
| Gordon and Betty Moore Foundation | Center for Integrative Study of Cell Regulation | Mary B Kennedy |
| Hicks Foundation | | Mary B Kennedy |
| Allen and Lenabelle Davis Foundation | Endowed Chair | Mary B Kennedy |
| National Institutes of Health | MH095095 | Mary B Kennedy |
| The Beckman Institute | Protein Expression Center | Jost Vielmetter |

The funders had no role in study design, data collection and interpretation, or the decision to submit the work for publication.

### Author contributions

WGW, TLM, Conception and design, Acquisition of data, Analysis and interpretation of data, Drafting or revising the article; LTS, RH, AI, MR, BDB, Acquisition of data, Analysis and interpretation of data; JV, Conception and design, Acquisition of data, Analysis and interpretation of data; MBK, Conception and design, Analysis and interpretation of data, Drafting or revising the article

### Author ORCIDs

Mary B Kennedy, http://orcid.org/0000-0003-1369-0525

### Ethics

Animal experimentation: This study was performed in strict accordance with the recommendations in the Guide for the Care and Use of Laboratory Animals of the National Institutes of Health. All of the animals were handled according to approved institutional animal care and use committee (IACUC) protocol (#1034-15G) of the California Institute of Technology.

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
