## [Decision Letter]

[Editors’ note: this article was originally rejected after discussions between the reviewers, but the authors were invited to resubmit after an appeal against the decision.]

Thank you for submitting your work entitled "Binding of synGAP to PSD-95 is regulated by phosphorylation and shapes the composition of the postsynaptic density" for consideration by *eLife*. Your article has been favorably evaluated by Eve Marder as the Senior Editor and three reviewers, one of whom, Leslie Griffith, is a member of our Board of Reviewing Editors.

We apologize for the delay in getting this decision to you, but it was partially caused by extensive back and forth discussions among the reviewers, during which several issues surfaced that were not necessarily highlighted in the initial independently prepared reviews. Based on these discussions and the individual reviews below, we regret to inform you that your work will not be considered further for publication in *eLife*.

While all the reviewers found this study interesting, there was a consensus that as it stands it is not suitable for *eLife*. This was a very difficult decision since the paper begins to address the much-needed quantitative measures of protein-protein interactions in the PSD (in this case of the PSD-95/SynGAP interaction). The other appeal of this paper was the clear explication of a model for synaptic expansion. This idea is in some senses is the major strength of the paper.

There were, however, several negative factors that outweighed these positives. The main issue is that the qualitative findings of Araki et al. provide the same bottom line as this paper, but with more physiological relevance. This is not, we would like to stress, just a matter of "publication date". While there are new quantitative data on the sites studied in Araki (these sites were originally identified by the Kennedy group) the focus is on a new site, S1283, identified in vitro. The paper fails to provide a clear integrated discussion of the roles of these sites. The S1123/S1093 sites seem likely to be the relevant "10 min sites" that are causing PSD-95 release in vitro in the S1283 mutant and it seems equally likely that S1283 is the "10 min" site that causes release in the S1123 mutants, making it unclear why the S1123/S1283 double mutant was not examined. This paper needs to make a compelling case as to why quantitative information about S1283, an in vitro phosphorylation site which has not been shown to have a role in vivo, gives added value to the story. Secondly, there is no discussion of how the non-PDZ ligand form of SynGAP is released by phosphorylation, something a more general model should include. Thirdly, the SynGAP/+ PSD data, while suggestive and supportive of the model, are not totally compelling due to the potential for developmental effects.

*Reviewer #1:*

This paper provides a beautifully logical and well-supported model for how activity can lead to the remodeling of the PSD and the insertion of AMPARs. The study demonstrates very convincingly that phosphorylation of SynGAP by CaMKII reduces its affinity for PSD-95 PDZ domains. The authors go on to posit that in vivo this decrease in PDZ binding frees up "slots" for other PSD-95-binding proteins to inhabit and allows the insertion of more AMPARs into the synapse. This idea has a certain appeal since it helps explain the initiation of the remodeling of the PSD that is caused by activity. The model is circumstantially supported by data from PSDs of WT and synGAP heterozygotes which show that the ratio of PSD-95 to SynGAP is decreased, but the ratio to other PDZ-binding partners is increased.

In general this is a very interesting, convincing and potentially important study. I have only two suggestions. The first is that given the ability of the authors to look at PSD-95 binding in a quantitative manner they should have filled in the gap between their biochemical data and the PSD immunoblots by showing with binding assays that SynGAP and other PDZ ligands actually compete and that phosphorylation of SynGAP alters its ability to compete. This would nicely close the loop.

The second is that the authors seem to take a very CaMKII-centric view of SynGAP phosphorylation. It clearly is the big dog with 10 mol/mol, but can other kinases cause similar losses of affinity for PSD-95? How about Cdk5 or PKA? Can they act independently or synergistically with CaMKII? Given that it appears that the affinity effect is "bulk" and not site-specific this may be quite biologically relevant. It at least merits discussion.

*Reviewer #2:*

This manuscript presents evidence that phosphorylation of SynGAP by CaMKII near its C-terminus decreases its binding to the PDZ domains of PSD-95. The authors identify S1283 as a critical phosphorylation site in this context. The data are largely solid. However, recent other work already showed that phosphorylation of SynGAP on S1108 and S1128 by CaMKII reduces its interaction with PSD-95 and is important for chemically induced LTP (Araki et al., 2015, as cited). The authors would have to discuss how their findings intersect with this recent other work in much more detail than they present right now. Most importantly, in the absence of any functional studies such as redistribution of postsynaptic proteins upon S1283 phosphorylation and its dependence on S1283 phosphorylation, the functional significance of this phosphorylation is not clear enough for publication in *eLife*.

A technical major issue is that pull-down data lack controls for non-specific binding of the various SynGAP polypeptides to resin, ideally with a non-specific tag. This is in part important because there is quite a bit of difference in Figure 4 for pull down of WT SynGAP after 0.5 vs. 10 min CaMKII pre-treatment. Could this be due differences between experiments, perhaps due to variations in non-specific binding to resin?In this context, the fact that the point mutations of SynGAP showed more binding than WT after 0.5 min CaMKII pretreatment could potentially be due to variation in non-specific binding and, if so, there might not be a difference between the 0.5 and 10 min CaMKII incubations in that respect.

*Reviewer #3:*

This is a well-written and clearly presented manuscript describing a novel mechanism by which synGAP regulates synaptic function. It is of high importance given the fact that mutations in synGAP in humans leads to non-syndromic intellectual disability and epileptic encephalopathy. Hence, understanding the mechanisms of synGAP action will help devise therapeutic strategies for these debilitating disorders.

While overall this is an exciting manuscript, I have several criticisms that need to be addressed.

1) As it stands, the manuscript reads as if all isoforms of synGAP bind PSD-95 through the PDZ-binding domain. However, this is true only for those isoforms containing the α1 c-terminal. The relative abundance of these protein isoforms relative to α2 (or other isoforms) is not well-described but both localize to synapses. A clear statement that the mechanism described in this manuscript applies to α1 containing isoforms only is warranted. Preferably, instead of referring to synGAP, the authors could refer to synGAP α1 throughout the manuscript. While these changes do not detract from the importance of the main findings of the manuscript, they highlight that many synGAP isoform exist at the synapse that do not contain the sequences necessary to bind PSD-95.

2) It is unclear how they derived the equation to determine the best fit of their data. In several of their panels, the curves do not appear to fit the data well at all. How were the constraints placed on the maxima and minima? For example, for much of their data, the levels of free r-synGAP never reach 0, despite high levels of PDZ concentrations. However, the curves all asymptote to 0. Hence, the equation does not appear to describe the data. Consideration needs to be given to alternative equation that better describe the data to allow accurate K_D_s to be determined.

[Editors’ note: what now follows is the decision letter after the authors submitted for further consideration.]

Thank you for submitting your article "Binding of synGAP to PSD-95 is regulated by phosphorylation and shapes the composition of the postsynaptic density" for consideration by *eLife*. Your article has been favorably evaluated by Eve Marder as the Senior Editor and two reviewers: Peter C. Kind (Reviewer #3) and Leslie C Griffith (Reviewer #1), who is a member of our Board of Reviewing Editors.

The reviewers have discussed the reviews with one another and the Reviewing Editor has drafted this decision to help you prepare a revised submission.

Summary:

This paper proposes a model for how activity-dependent synaptic reorganization occurs. The authors demonstrate that CaMKII phosphorylation of synGAP reduces its affinity for the PDZ domains of PSD-95, thus making these sites available for other proteins. This role for synGAP in regulating PSD composition is independent of it GTPase activity and changes the potential outcomes of both synGAP action in normally developing neurons as well as in the human pathology associated with SYNGAP haploinsufficiency. The biochemical findings are tightly integrated into a model of PSD plasticity that provides a new way of looking at how synapses reorganize. While not all the last details have been worked out, this is a substantially new way of looking at the PSD that should have a significant influence on the field.

Essential revisions:

The authors should deal with the minor discussion points outlined below.

1) While the data showing an increase in binding of TARPs, LRRTM2 and NLGN2 is consistent with the model proposed, I do not think the authors should be quite so dismissive of the potential of developmental compensation as a contributing factor. Identifying direct vs. indirect effects of a given mutation is always difficult and compensation can happen at all levels, from molecular to circuit. *SynGAP^+/-^* neurons show numerous physiological in intrinsic and circuit properties, plasticity, morphology etc. How each of these relate directly to the reduction in synGAP is not known. Since the excitability and activity of neurons is altered, so to will the activity-dependent gene expression. Hence, changes in protein expression and localisation may also arise as a secondary effect of changes in activity across development. Therefore, while their data certainly are consistent with their model and their conclusions may even be the most parsimonious explanation for their data since TARPs, LRRTM2 and NLGN2 are all affected, a critical discussion of other possibilities seems warranted.

2) The McMahon et al. manuscript did not make any conclusions about the relative abundance of α1 and α2 isoforms because different antibodies with different binding affinities were used. That said, it is clear that α1 containing synGAPs are abundantly expressed at synapses and that the mechanism proposed here is an important regulator of synGAP function.

3) The Barnett et al. manuscript did not show that, "isoforms of synGAP that do not contain the PDZ ligand are, nevertheless, bound to the PSD." as stated in the Discussion. Instead they showed that in PSD-95 mutants, synGAP still associated with the PSD using a pansynGAP antibody. They did not distinguish whether these were non-PDZ binding isoforms or α1 containing isoforms that use and non PSD-95 strategy for associating with the PSD. However, McMahon et al. presented mass spec data showing α2 containing isoforms are present at excitatory synapses.

4) A clear comparison of the different phosphorylation sites between the Araki et al. and this manuscript would be useful. As far as I can tell, both the Araki et al. and this manuscript are referring to the rat sequence. However, the Araki paper is using the numbering of the A isoform and the Kennedy is using the α1 isoform. These can be seen in the McMahon et al., Supplemental Table 1. They encode different N-terminal peptides that differ by 15 amino acids and this leads to the differences in numbering. Is this correct?

5) It does not appear my original query on statistics was addressed. So I restate it below. Apologies if I missed any clarification.

In general the statistical methods appear appropriate. One clarification, for Figure 9 could the authors please confirms that n=number of animals rather than number of PSD preps or number of lanes. I believe this is the case, but it is ambiguous.

---

## [Author Response]

[Editors’ note: the author responses to the first round of peer review follow.]

*While all the reviewers found this study interesting, there was a consensus that as it stands it is not suitable for eLife. This was a very difficult decision since the paper begins to address the much-needed quantitative measures of protein-protein interactions in the PSD (in this case of the PSD-95/SynGAP interaction). The other appeal of this paper was the clear explication of a model for synaptic expansion. This idea is in some senses is the major strength of the paper.*

*There were, however, several negative factors that outweighed these positives. The main issue is that the qualitative findings of Araki et al. provide the same bottom line as this paper, but with more physiological relevance. This is not, we would like to stress, just a matter of "publication date".*

This statement is demonstrably false. The conclusions of the Araki paper were completely different from ours and offered a different model. Indeed most of the experiments in our manuscript are entirely different. The summary from the Araki paper taken from the Neuron PDF of the paper states: “In Brief:

Araki et al. discovered that CaMKII- dependent phosphorylation of SynGAP induces its dispersion from synaptic spines leading to activation of small G proteins during LTP. These results define a key missing link between CaMKII and small G protein activation.”

There is no mention made in the Araki paper of the possibility that “release” of synGAP is important for reducing competition for binding to PDZ domains of PSD-95. Nor are there any experiments dealing with that issue.

Our study adds the following new information not published previously (in addition to all of the quantitative data):

A) Most importantly, we provide a new model for a distinct function of synGAP by recognizing the consequences of the fact that synGAP is an unusually abundant protein in the PSD, nearly as abundant as PSD-95 itself. Our reading of the data of McMahon et al. on synGAP isoforms is that the synGAP α1 isoform makes up approximately half of the synGAP in a mouse hippocampal homogenate (at the very least 30%). This fact predicts that synGAP could occupy as many as 15% of all of the PDZ domains on PSD-95. This means that a reduction in affinity of synGAP for the PDZ domains would necessarily make a relatively large number of PDZ domains available for binding to other proteins. This has not been stated in publications previously and this simple deduction has not been made in any other paper. The model is a significant advance on its own. As the editors suggested, we have rearranged the paper to introduce this model thoroughly in the introduction, and we have emphasized it in the title.

B) We make an experimental prediction that is improbable unless the mechanism we propose is operating, and we validate that prediction with data. Proof of our model will require many experiments, but Occam’s razor indicates that our model is the most parsimonious explanation of our data.

C) We show that cumulative phosphorylation of synGAP at several sites in the C-terminus reduces the affinity of synGAP for ALL THREE of the PDZ domains on PSD-95.

Finally, we make the comment that, though it is widely stated that experiments involving expression of fluorescent proteins and live imaging are “more physiological” than biochemical studies, we do not share this view. Such experiments cannot control for over- and under expression of fluorescent proteins and resulting artifacts arising from disrupted kinetics. They are useful to provide qualitative data about behavior of individual molecules in vivo. However, they are prone to erroneous interpretation and must be understood in light of the biochemical characteristics of the proteins studied. We believe that progress in understanding molecular mechanisms within neurons has been damaged by over-reliance on studies of individual fluorescent proteins in vivo by imaging.

*While there are new quantitative data on the sites studied in Araki (these sites were originally identified by the Kennedy group) the focus is on a new site, S1283,* identified *in vitro.*

The focus of our study was not and is not on identifying a new site. We agree with the reviewers that if the only addition that our work made to the qualitative Araki study is the identification of one possible new site, S1283, the paper would not be suitable for *eLife*. However, nothing in our paper indicated that this finding as its major focus. In our view, the identification of the additional site simply cleans up and clarifies the number of sites that can be involved in regulating binding of synGAP to PDZ domains. It is a detail.

*The paper fails to provide a clear integrated discussion of the roles of these sites. The S1123/S1093 sites seem likely to be the relevant "10 min sites" that are causing PSD-95 release in vitro in the S1283 mutant and it seems equally likely that S1283 is the "10 min" site that causes release in the S1123 mutants, making it unclear why the S1123/S1283 double mutant was not examined. This paper needs to make a compelling case as to why quantitative information about S1283, an in vitro phosphorylation site which has not been shown to have a role in vivo, gives added value to the story.*

As we stated above, our goal was not and is not to provide an integrated discussion of the roles of sites S1123 and S1283 in synGAP. We have added studies of additional multiple mutations that included site S1283. We now conclude that both S1123 and S1283 can affect binding affinity for PDZ domains, but the effect of S1283 seems to be somewhat more potent. We show in two ways that cumulative phosphorylation of other, more slowly phosphorylated sites by both CaMKII and PLK2 can also result in a decrease in affinity. This data shows that synGAP conforms to a model of regulation stressed by Holt et al. (cited in the manuscript) which states that phosphorylation often involves disruption or enhancement of protein-protein interactions by addition of multiple negative charges, often in an otherwise disordered region. We do not believe that it is accurate to imagine that particular sites in synGAP have unique functions in regulating association with PDZ domains.

*Secondly, there is no discussion of how the non-PDZ ligand form of SynGAP is released by phosphorylation, something a more general model should include. Thirdly, the SynGAP/+ PSD data, while suggestive and supportive of the model, are not totally compelling due to the potential for developmental effects.*

The reviewers are correct that we did not discuss explicitly how β or α2 isoforms that don’t have PDZ binding sites might be released from the PSD. A paper by Yang et al. (2013) in PLoS One, from the Dosemeci/Reese group shows movement of α2 isoforms away from the membrane. It is an EM study showing that synGAP α2 and synGAP α1 both move from inside the PSD at ~30 nm from the membrane to what they call the “pallium” of the PSD, at a median of ~60 nm from the membrane. The movement is evident 10 min after NMDA treatment, and the movement is sensitive to inhibition by a peptide inhibitor of CaMKII. This result suggests that phosphorylation can indeed affect the micro-distribution of synGAP α2 similarly to synGAP α1. This is an issue that we are interested in examining, but it is not the focus of this paper to provide a “general” model of the functions of all synGAP isoforms. The existence of additional isoforms does not directly impinge on the hypothesis that is the central focus of our manuscript. We do not believe that we should be required to figure out every aspect of the function of synGAP in order to publish this paper in *eLife*.

We have added the suffix α1 at all places in our manuscript where it is appropriate, as suggested by reviewer 3; and we have modified the Discussion to take into account that the α1 subunit is only approx. half of the total synGAP. Even a conservative interpretation of the data in McMahon et al. would suggest that at least 30% of the total synGAP is an α1 isoform. These numbers don’t alter the predictions of the model substantially.

We disagree with the statement of the reviewers that “…the *SynGAP^-/+^* PSD data, while suggestive and supportive of the model, are not totally compelling due to the potential for developmental effects.” SynGAP begins to be expressed as synapses begin forming in the brain. Newborns with complete deletion of synGAP appear normal, but die a few days after birth with movement defects and apparent seizure activity, indicating a defect as synapses are forming. Such developmental effects of mutations arise for molecular reasons. There is no evidence of morphological changes in synGAP heterozygous animals, nor in individuals with synGAP haploinsufficiency, that might lead to a wiring defect that could produce the result we report. The hypothesis we have put forward is by far the most parsimonious explanation for the change in composition of forebrain PSDs in the *synGAP^+/-^* animals. Indeed, the quantitative nature of the changes is difficult to explain in any other ways. They cannot be explained by a two-fold reduction in RasGAP and RapGAP activity. SynGAP produces a much larger fold increase in GTPase activity of Rap than of Ras (Walkup et al., 2015 and references therein). Therefore, loss of half of the GAP activity would be predicted to increase the level of active Rap even more than that of active Ras. Thus, one would predict a relative increase in endocytosis of AMPARs compared to exocytosis. That would result in fewer AMPAR binding proteins in the PSD; not more, as we observe. Beyond that, the competitive role for synGAP binding explains several previously unexplained observations in the literature. Many of these were detailed in our discussion. For now, Occam’s razor strongly favors our hypothesis. We point out that many experiments will be necessary to “prove” this model beyond a doubt. However, the experiment we present is a strong favorable one given that it matches the prediction of the model.

*Reviewer #1:*

*This paper provides a beautifully logical and well-supported model for how activity can lead to the remodeling of the PSD and the insertion of AMPARs. The study demonstrates very convincingly that phosphorylation of SynGAP by CaMKII reduces its affinity for PSD-95 PDZ domains. The authors go on to posit that in vivo this decrease in PDZ binding frees up "slots" for other PSD-95-binding proteins to inhabit and allows the insertion of more AMPARs into the synapse. This idea has a certain appeal since it helps explain the initiation of the remodeling of the PSD that is caused by activity. The model is circumstantially supported by data from PSDs of WT and synGAP heterozygotes which show that the ratio of PSD-95 to SynGAP is decreased, but the ratio to other PDZ-binding partners is increased.*

*In general this is a very interesting, convincing and potentially important study. I have only two suggestions. The first is that given the ability of the authors to look at PSD-95 binding in a quantitative manner they should have filled in the gap between their biochemical data and the PSD immunoblots by showing with binding assays that SynGAP and other PDZ ligands actually compete and that phosphorylation of SynGAP alters its ability to compete. This would nicely close the loop.*

We are interested in doing such a study, but it is far more complicated and tricky than the reviewer imagines. The appropriate controls and a thorough job would require an additional paper.

*The second is that the authors seem to take a very CaMKII-centric view of SynGAP phosphorylation. It clearly is the big dog with 10 mol/mol, but can other kinases cause similar losses of affinity for PSD-95? How about Cdk5 or PKA? Can they act independently or synergistically with CaMKII? Given that it appears that the affinity effect is "bulk" and not site-specific this may be quite biologically relevant. It at least merits discussion.*

We have added experiments with CDK5 and PLK2, both of which phosphorylate synGAP, but more slowly than CaMKII. CDK5 phosphorylation has no effect on affinity; whereas PLK2 phosphorylation decreases affinity less than phosphorylation by CaMKII. Its effect adds to the conclusion that “bulk phosphorylation” can have a greater effect than single sites.

*Reviewer #2:*

*This manuscript presents evidence that phosphorylation of SynGAP by CaMKII near its C-terminus decreases its binding to the PDZ domains of PSD-95. The authors identify S1283 as a critical phosphorylation site in this context. The data are largely solid. However, recent other work already showed that phosphorylation of SynGAP on S1108 and S1128 by CaMKII reduces its interaction with PSD-95 and is important for chemically induced LTP (Araki et al., 2015, as cited). The authors would have to discuss how their findings intersect with this recent other work in much more detail than they present right now. Most importantly, in the absence of any functional studies such as redistribution of postsynaptic proteins upon S1283 phosphorylation and its dependence on S1283 phosphorylation, the functional significance of this phosphorylation is not clear enough for publication in eLife.*

*A technical major issue is that pull-down data lack controls for non-specific binding of the various SynGAP polypeptides to resin, ideally with a non-specific tag. This is in part important because there is quite a bit of difference in Figure 4 for pull down of WT SynGAP after 0.5 vs. 10 min CaMKII pre-treatment. Could this be due differences between experiments, perhaps due to variations in non-specific binding to resin? In this context, the fact that the point mutations of SynGAP showed more binding than WT after 0.5 min CaMKII pretreatment could potentially be due to variation in non-specific binding and, if so, there might not be a difference between the 0.5 and 10 min CaMKII incubations in that respect.*

There is no detectable non-specific binding of synGAP protein to the affinity resin under the conditions of the bead binding assay. We now state this in the manuscript.

*Reviewer #3:*

*This is a well-written and clearly presented manuscript describing a novel mechanism by which synGAP regulates synaptic function. It is of high importance given the fact that mutations in synGAP in humans leads to non-syndromic intellectual disability and epileptic encephalopathy. Hence, understanding the mechanisms of synGAP action will help devise therapeutic strategies for these debilitating disorders.*

*While overall this is an exciting manuscript, I have several criticisms that need to be addressed.*

*1) As it stands, the manuscript reads as if all isoforms of synGAP bind PSD-95 through the PDZ-binding domain. However, this is true only for those isoforms containing the α1 c-terminal. The relative abundance of these protein isoforms relative to α 2 (or other isoforms) is not well-described but both localize to synapses. A clear statement that the mechanism described in this manuscript applies to α1 containing isoforms only is warranted. Preferably, instead of referring to synGAP, the authors could refer to synGAPα1 throughout the manuscript. While these changes do not detract from the importance of the main findings of the manuscript, they highlight that many synGAP isoform exist at the synapse that do not contain the sequences necessary to bind PSD-95.*

As suggested, we now refer to synGAP-α1 throughout the manuscript and state that the model applies to that isoform.

*2) It is unclear how they derived the equation to determine the best fit of their data. In several of their panels, the curves do not appear to fit the data well at all. How were the constraints placed on the maxima and minima? For example, for much of their data, the levels of free r-synGAP never reach 0, despite high levels of PDZ concentrations. However, the curves all asymptote to 0. Hence, the equation does not appear to describe the data. Consideration needs to be given to alternative equation that better describe the data to allow accurate* K_D_*s to be determined.*

The equation was derived in earlier papers that we cited and is described in the Biacore handbook. We now state in the Methods and Results that the equation describes an ideal situation in which there is one binding site between ligand and receptor, and in which there is no non-specific binding. We now indicate why the levels of free r-synGAP-α1 do not appear to reach zero (binding between free soluble PDZ domain and PDZ domains on the chip at high concentration). Despite this deviation, the data allow a sufficient fit to calculate a reliable K_D_ for a single site. We also explain why we use the term apparent K_D_ for the numbers derived for constructs with more than one PDZ domain. Because the two or three PDZ domains have similar, but not identical, affinities the curve will not be fit exactly. There will be multiple solutions to equations derived assuming more than one binding site. Simply put, there are not good ways to use bulk methods to find the affinities of individual sites in a molecule with more than one binding site. The “apparent” K_D_ is, nonetheless, a useful description of the summed binding properties of the constructs with multiple sites.

[Editors’ note: the author responses to the re-review follow.]

*The authors should deal with the minor discussion points outlined below.*

*1) While the data showing an increase in binding of TARPs, LRRTM2 and NLGN2 is consistent with the model proposed, I do not think the authors should be quite so dismissive of the potential of developmental compensation as a contributing factor. Identifying direct vs. indirect effects of a given mutation is always difficult and compensation can happen at all levels, from molecular to circuit. SynGAP^+/-^ neurons show numerous physiological in intrinsic and circuit properties, plasticity, morphology etc. How each of these relate directly to the reduction in synGAP is not known. Since the excitability and activity of neurons is altered, so to will the activity-dependent gene expression. Hence, changes in protein expression and localisation may also arise as a secondary effect of changes in activity across development. Therefore, while their data certainly are consistent with their model and their conclusions may even be the most parsimonious explanation for their data since TARPs, LRRTM2 and NLGN2 are all affected, a critical discussion of other possibilities seems warranted.*

We have added an additional paragraph to the Introduction describing yet another experiment from the literature that is explained by the proposed model. We did this in order to refer to it in a paragraph that we added to the Discussion. In this latter paragraph we discuss the possibility of other effects during development, and we explain why we believe that Occam’s razor favors our model at this time.

*2) The McMahon et al. manuscript did not make any conclusions about the relative abundance of α1 and α2 isoforms because different antibodies with different binding affinities were used. That said, it is clear that α1 containing synGAPs are abundantly expressed at synapses and that the mechanism proposed here is an important regulator of synGAP function.*

We made our estimate based on a comparison of the relative ratios of the major α1 and α2 isoforms revealed by the two isoform-specific antibodies compared to the pan-synGAP antibody. This estimate would not depend strongly on the affinities of the different antibodies. Nonetheless, it is a soft estimate and we have removed the citation of McMahon et al. and simply stated the assumption that α1 makes up 30-50% of the synaptic synGAP.

*3) The Barnett et al. manuscript did not show that, "isoforms of synGAP that do not contain the PDZ ligand are, nevertheless, bound to the PSD." as stated in the Discussion. Instead they showed that in PSD-95 mutants, synGAP still associated with the PSD using a pansynGAP antibody. They did not distinguish whether these were non-PDZ binding isoforms or α1 containing isoforms that use and non PSD-95 strategy for associating with the PSD. However, McMahon et al. presented mass spec data showing α2 containing isoforms are present at excitatory synapses.*

We have changed the references for this point and removed the reference to the Barnett paper.

*4) A clear comparison of the different phosphorylation sites between the Araki et al. and this manuscript would be useful. As far as I can tell, both the Araki et al. and this manuscript are referring to the rat sequence. However, the Araki paper is using the numbering of the A isoform and the Kennedy is using the α1 isoform. These can be seen in the McMahon et al., Supplemental Table 1. They encode different N-terminal peptides that differ by 15 amino acids and this leads to the differences in numbering. Is this correct?*

We have added a key to Figure 2 comparing the numbering of sites in our papers, the Araki paper, and also the paper by Lee et al. on Plk2 sites. Lee et al. use numbers from the β isoform.

*5) It does not appear my original query on statistics was addressed. So I restate it below. Apologies if I missed any clarification.*

*In general the statistical methods appear appropriate. One clarification, for Figure 9 could the authors please confirms that n=number of animals rather than number of PSD preps or number of lanes. I believe this is the case, but it is ambiguous.*

We did add what we thought was a clarification to the legend of Figure 11. However, we have now made the reference to the units of n more explicit. We note that n does *not* refer to the number of animals here. It refers to the individual measurements of protein ratios from each lane of gels. The number of animals is 6 wild type and 6 het in all cases. It is not customary or usually necessary to make biochemical measurements on individual animals. In behavioral experiments the variation from animal to animal can be large because of the many brain areas involved in expressing behaviors, and the difficulty of controlling context perfectly. Biochemical variation from animal to animal is generally much smaller, and measurements are made as averages of preparations from brain tissue pooled from several animals. In this case, the technical replicates are important because the variation in measurements from individual lanes is high. This variation results from minor differences in loading, immunoblot background, shapes of lanes, small variations in amounts of protein transferred during blotting, etc.